# Cooperative Bargaining Games Without Utilities: Mediated Solutions from Direction Oracles

**Kushagra Gupta**[*][†]
The University of Texas at Austin
kushagrag@utexas.edu

**Surya Murthy**[*]
The University of Texas at Austin
surya.murthy@utexas.edu

**Mustafa O. Karabag**
The University of Texas at Austin
karabag@utexas.edu

**Ufuk Topcu**
The University of Texas at Austin
utopcu@utexas.edu

**David Fridovich-Keil**
The University of Texas at Austin
dfk@utexas.edu

## Abstract

Cooperative bargaining games are widely used to model resource allocation and conflict resolution. Traditional solutions assume the mediator can access agents' utility function values and gradients. However, there is an increasing number of settings, such as human-AI interactions, where utility values may be inaccessible or incomparable due to unknown, nonaffine transformations. To model such settings, we consider that the mediator has access only to agents' *most preferred directions*—normalized utility gradients in the decision space. To this end, we propose a cooperative bargaining algorithm where a mediator has access to only the direction oracle of each agent. We prove that unlike popular approaches such as the Nash and Kalai-Smorodinsky bargaining solutions, our approach is invariant to monotonic nonaffine transformations, and that under strong convexity and smoothness assumptions, this approach enjoys global asymptotic convergence to Pareto stationary solutions. Moreover, we show that the bargaining solutions found by our algorithm also satisfy the axioms of symmetry and (under slightly stronger conditions) independence of irrelevant alternatives, which are popular in the literature. Finally, we conduct experiments in two domains, multi-agent formation assignment and mediated stock portfolio allocation, which validate these theoretical results.
Project Website: https://kaugsrha.github.io/dibs-on-neurips.

## 1 Introduction

We consider the problem of centralized, cooperative multi-agent bargaining where a mediator has access to only each agent's *most preferred direction* to move in the decision space, and does not know the agents' underlying utility functions. Over the past eighty years, a variety of celebrated bargaining solution concepts have been introduced [26]; however, most require the mediator to have access to the explicit utility values and gradients for each agent, e.g., the Nash [19] and Kalai-Smorodinsky [9] bargaining solutions (NBS and KSBS, respectively). Unfortunately, these existing bargaining solutions cannot cater to an increasing number of important settings.

---

[*]Equal contribution.
[†]Corresponding author.

39th Conference on Neural Information Processing Systems (NeurIPS 2025).

An example of this is when mediators do not have access to agents' underlying utilities (or their gradients) in human-AI interactions. Humans or AI-based proxies (e.g., language models) may have a clear idea of a desired bargaining outcome, but have difficulty providing a numerical value of the utility associated to arbitrary outcomes. Even if a utility is available, they may not wish to share that information for *privacy* reasons.

A separate, but equally important issue arises when agents' utilities are not directly comparable, e.g. because they are scaled in different and potentially nonaffine ways. Traditional bargaining solutions like NBS and KSBS are invariant to affine transformations in agents' utilities; however, they can lose this invariance under *nonaffine* transformations [26] (such as those arising in prospect-theoretic models [28, 30], or which are represented by neural network based AI-proxies [1]) and yield solutions which favor one agent disproportionately. In such scenarios, despite utilities being incomparable due to different nonaffine scalings, the notion of the agents' most preferred directions remains intact.

Such settings motivate the need for an approach to bargaining in scenarios where the mediator only has access to a direction oracle which provides the most preferred direction (i.e., the normalized utility gradient) in the decision space for each agent. Inspired by this, we address the following questions:

1. *Is it possible to solve for existing bargaining solution concepts which are based on utility values in the setting where the mediator has access to only agents' most preferred directions, and their utilities?*
   **Contribution 1.** We show that no algorithm with mediator access to only direction oracles can find the Nash or Kalai-Smorodinsky bargaining solutions for all bargaining games satisfying standard assumptions in the literature.

2. *If not, can we develop a solution concept for the direction oracle setting where the mediator effectively balances every agent's interests, and is invariant to preference-preserving non-affine transformations of (potentially unknown) agent utilities?*
   **Contribution 2.** We propose **Di**rection-based **B**argaining **S**olution (DiBS), an iterative algorithm which uses only direction oracles, and reasons about every agent's distance from their preferred state throughout the bargaining process. We show that under standard assumptions common in the literature, the solution found by DiBS satisfies the following axioms:
   (a) Pareto stationarity (a necessary condition for Pareto optimality, and also sufficient under slightly stronger assumptions).
   (b) Invariance to strictly increasing monotonic *nonaffine* transformations.
   (c) Symmetry, and under slightly stronger conditions, independence of irrelevant alternatives.

3. *Can we have convergence guarantees to reliably find bargaining solutions which employ only direction oracles?*
   **Contribution 3.** We prove that under standard assumptions, fixed points of DiBS exist, all its fixed points are Pareto stationary points, and that assuming that the (hidden) agent costs (i.e., negative utilities) are strongly convex and smooth, DiBS enjoys global asymptotic convergence guarantees to the set of its fixed points.

We note that access to the direction oracle can be easily achieved in practice by giving the mediator access to a minimal information *comparison oracle* for each agent – at some state in the bargaining process, the mediator proposes new states to every agent, who responds either "yes", "no" or "indifferent", depending on whether the agent prefers the new state more than the current state. Forming estimates of agents' most preferred directions (i.e., normalized utility gradients) up to arbitrary accuracy from only a (potentially noisy) comparison oracle is a well-studied problem, with many established algorithms [6, 10, 13, 24, 25, 34]. Of course, when agents' utilities are available, the mediator can simply compute their normalized gradients; these will be comparable across agents even if their utilities are scaled in different nonaffine fashions.

## 2   On existing bargaining solutions and related work

**Notation.** We denote a vector in bold (e.g., $\mathbf{x}$). We denote $[N] := \{1, 2, \ldots, N\}$. For $\mathbf{x}, \mathbf{y} \in \mathbb{R}^n$, inequalities apply elementwise. For some process involving iterations of a vector $\mathbf{x}$, we denote the $k^{\text{th}}$ iteration as $\mathbf{x}_k$.

**Bargaining game with known utilities.** Consider a game with $N$ agents, the $i^{\text{th}}$ of which has differentiable cost (i.e., negative utility) function $\ell^i(\mathbf{x}) : \mathbb{R}^n \rightarrow \mathbb{R}$, where $\mathbf{x} \in \mathcal{S} \subset \mathbb{R}^n$ denotes the state of the game. Each agent wants to minimize their cost, which corresponds to moving to a set of preferred states in $\mathcal{S}$, $\mathbf{x}^{*,i} \in \arg\min_{\mathbf{x} \in \mathcal{S}} \ell^i(\mathbf{x})$ and we assume that $\mathbf{x}^{*,i}$ exists. The mediator must conduct a bargaining process to output a *bargaining solution* state $\mathbf{x}^\dagger$. Every agent is incentivized to participate in the bargaining process by being assigned a disagreement penalty of $d^i \in \mathbb{R}$ in case no bargaining solution is agreed upon. Let $\boldsymbol{\ell}(\mathbf{x}) := \left[\ell^1(\mathbf{x}), \cdots, \ell^N(\mathbf{x})\right]$, $\mathbf{d} := \left[d^1, \cdots, d^N\right]$, and $\mathcal{L} = \{\boldsymbol{\ell}(\mathbf{x}) | \mathbf{x} \in \mathcal{S}\}$. It is assumed that $\mathcal{S} \cap \{\mathbf{x} \in \mathbb{R}^n : \mathbf{d} > \boldsymbol{\ell}(\mathbf{x})\}$ is non-empty. We denote such a bargaining game by $\mathcal{B}_{\mathcal{S}}(\boldsymbol{\ell}, \mathbf{d})$, and any process finding a bargaining solution to $\mathcal{B}_{\mathcal{S}}(\boldsymbol{\ell}, \mathbf{d})$ must output a solution state $\mathbf{x}^\dagger$ and the corresponding agent costs $\boldsymbol{\ell}(\mathbf{x}^\dagger)$. A detailed description of bargaining games can be found in existing literature, cf. [18, 22, 26].

It is widely recognized that there is no one correct approach to solve a bargaining problem. Nash proposed a set of desirable *axioms* (given below) that a bargaining solution should satisfy [19, 22], showing his Nash Bargaining Solution (NBS) can be found by optimizing the product of agents' utilities:

**Axiom 1** (Weak Pareto Optimality). *A state $\mathbf{x}^\dagger$ is weakly Pareto optimal if there does not exist a state $\mathbf{y} \in \mathcal{S}$ such that $\boldsymbol{\ell}(\mathbf{x}^\dagger) > \boldsymbol{\ell}(\mathbf{y})$, $\mathbf{y} \neq \mathbf{x}^\dagger$.*

**Axiom 2** (Symmetry). *The bargaining solution is invariant to permuting the agents' order.*

**Axiom 3** (Invariance to Affine Transformations). *Applying an affine transformation $h^i(l) = a^i l + b^i, l \in \mathbb{R}, a^i \in \mathbb{R}_+, b^i \in \mathbb{R}$ to the $i^{th}$ agent's cost does not change the bargaining solution state.*

**Axiom 4** (Independence of Irrelevant Alternatives). *If a bargaining problem $\mathcal{B}_{\mathcal{S}}(\boldsymbol{\ell}, \mathbf{d})$ has solution state $\mathbf{x}^\dagger \in \mathcal{S}'$ with $\mathcal{S}' \subset \mathcal{S}$ then the bargaining problem $\mathcal{B}_{\mathcal{S}'}(\boldsymbol{\ell}, \mathbf{d})$ also has solution state $\mathbf{x}^\dagger$.*

The following assumptions are standard in the bargaining literature, c.f., [22, 26].

**Assumption 1.** The state space $\mathcal{S}$ is convex, and the set of Pareto points lies in interior of $\mathcal{S}$.

**Assumption 2.** The agent cost $\ell^i$ is twice-differentiable and convex.

We remark that no axiom is universally accepted in the literature. For example, a popular and well-studied bargaining solution that does away with Axiom 4 is the Kalai-Smorodinsky bargaining solution (KSBS) [9]. The KSBS satisfies Axioms 1-3 in addition to the axiom of individual monotonicity, which states that for two bargaining problems $\mathcal{B}_{\mathcal{S}}(\boldsymbol{\ell}, \mathbf{d})$ and $\mathcal{B}_{\mathcal{S}'}(\boldsymbol{\ell}, \mathbf{d})$, if $\mathcal{S}' \subseteq \mathcal{S}$ and $\arg\min_{\mathbf{x} \in \mathcal{S}} \ell^i(\mathbf{x}) = \arg\min_{\mathbf{x} \in \mathcal{S}'} \ell^i(\mathbf{x})$, then $\ell^i(\mathbf{x}_{\mathcal{S}}^\dagger) \leq \ell^i(\mathbf{x}_{\mathcal{S}'}^\dagger)$, where $\mathbf{x}_{\mathcal{S}}^\dagger, \mathbf{x}_{\mathcal{S}'}^\dagger$ denote the solutions to $\mathcal{B}_{\mathcal{S}}(\boldsymbol{\ell}, \mathbf{d})$ and $\mathcal{B}_{\mathcal{S}'}(\boldsymbol{\ell}, \mathbf{d})$ respectively.

Numerous other bargaining solutions exist which relax some combination of Axioms 1-4 and satisfy other axioms [22, 26, 29]. However, for the purposes of our discussion, it is sufficient to focus on the NBS and KSBS to illustrate our arguments. Formally, the NBS optimizes the product of agent utilities (negative costs), and consists of the iterates (for $k \geq 0$ and some appropriate step sizes $\alpha_k > 0$)

$$\mathbf{x}_{k+1} = \mathbf{x}_k - \alpha_k \sum_{i=1}^{N} \frac{\nabla \ell^i(\mathbf{x}_k)}{d^i - \ell^i(\mathbf{x}_k)}, \tag{NBS}$$

while the Kalai-Smorodinsky bargaining solution has a geometric solution to the bargaining problem, seeking to equalize the proportional cost benefits of every agent, i.e., finding a state $\mathbf{x}_{\text{KSBS}}$ such that

$$\frac{d^i - \ell^i(\mathbf{x}_{\text{KSBS}})}{d^i - \ell^i(\mathbf{x}^{*,i})} = \frac{d^j - \ell^j(\mathbf{x}_{\text{KSBS}})}{d^j - \ell^j(\mathbf{x}^{*,j})} \; \forall \, i, j, \in [N] \tag{KSBS}$$

A condition which is closely related to Pareto optimality, given in Axiom 1 and applicable when agent costs are smooth, is Pareto *stationarity*.

**Definition 1** (Pareto Stationarity). For a bargaining game as defined above, $\mathbf{x}_{\text{PS}} \in \mathcal{S}$ is said to be Pareto stationary if zero is a convex combination of agents' cost gradients at $\mathbf{x}_{\text{PS}}$, i.e., $\exists \, \beta^i \geq 0, i = 1, \ldots, N$, such that $\sum_{i=1}^{N} \beta^i \nabla \ell^i(\mathbf{x}_{\text{PS}}) = 0$ and $\sum_{i=1}^{N} \beta^i = 1$.

Pareto stationarity is a necessary condition for Pareto optimality [15, 23], and a sufficient condition for *strong* Pareto optimality (given below) when agent costs are strictly convex and twice differentiable [8, 23]. Pareto stationarity is a first-order characterization and acts as a useful alternative to Axiom

1, because it is often difficult to check if a point is Pareto optimal in many bargaining settings. As such, many works addressing bargaining in challenging non-conventional settings seek to satisfy Pareto stationarity [20, 32, 33]. Henceforth, we will also consider Pareto stationarity in the bargaining solution we introduce. Under an additional assumption of strict convexity, any weakly Pareto optimal point found by NBS is also strongly Pareto optimal, which is defined in the axiom below.

**Axiom 5** (Strong Pareto Optimality). *A state $\mathbf{x}^\dagger$ is strongly Pareto optimal if $\forall\, i \in [N], \ell^i(\mathbf{x}^\dagger) > \ell^i(\mathbf{y}) \implies \exists\, j \in [N]$ such that $\ell^j(\mathbf{x}^\dagger) < \ell^j(\mathbf{y})$.*

## 2.1 Limitations of existing bargaining solutions

We first provide a formal definition for the direction oracle.

**Definition 2** (Direction Oracle). A direction oracle for agent $i$, $\mathcal{O}_\ell^{\mathcal{D},i}(\mathbf{x})$, gives the most preferred direction for agent $i$ at a state $\mathbf{x}$, given by

$$\mathcal{O}_\ell^{\mathcal{D},i}(\mathbf{x}) = \begin{cases} -\frac{\nabla \ell^i(\mathbf{x})}{\|\nabla \ell^i(\mathbf{x})\|_2}, & \mathbf{x} \neq \mathbf{x}^{*,i} \\ 0, & \mathbf{x} = \mathbf{x}^{*,i} \end{cases}. \tag{1}$$

**Sensitivity to nonaffine scaling.** Although utility-value based bargaining solutions like NBS and KSBS are robust to affine utility transformations, they remain adversely susceptible to more general monotonic transformations. Such nonaffine transformations may occur due to agents reporting exaggerated scaled utilities in order to bias the bargaining solution, or unintentionally when trying to model preferences as a numerical utility from data involving comparisons, like in reinforcement learning from human feedback (RLHF) [11, 31]. Another source of such transformations inadvertently appearing is when agents have utilities which fit hand-tailored reward functions representing preferences; this is especially common when deploying reinforcement learning in "sparse" reward scenarios, where a crafted dense reward is used as a proxy for sparse high-level preferences [16, 27]. While such transformations corrupt utility values and change bargaining solutions, they still roughly preserve agent preferences; bargaining solutions employing only direction oracles should remain robust to such nonaffine monotonic transformations (this will be proved in Section 3).

**Existing bargaining solutions cannot be found with direction oracles.** Most bargaining solutions, including NBS and KSBS, require the mediator to have access to agent costs/utilities. Intuitively, this is explained by the fact that these existing bargaining solutions require reasoning about agents' benefits in the cost space. For example, implementing the NBS solution requires knowledge of both agent cost values and gradients. However, when the mediator has access to only the direction oracles $\mathcal{O}_\ell^{\mathcal{D},i}, i = 1, \ldots, N$, the agents' cost values and gradient magnitudes are not available. This lack of information makes it impossible to find points which satisfy the NBS and KSBS solution concepts (even if they exist). This result is formalized in Proposition 1 (proof in Appendix A.1).

**Proposition 1** (Inadequacy of NBS and KSBS for the direction oracle). *There does not exist any bargaining algorithm in which a mediator with access to only direction oracles $\mathcal{O}_\ell^{\mathcal{D},i}$ can find the Nash or the Kalai-Smorodinsky bargaining solutions for all problems satisfying Assumptions 1-2.*

Proposition 1 establishes that existing bargaining solutions require reasoning about quantities which are inaccessible in the direction oracle setting. The proof of Proposition 1, given in Appendix A.1, exploits the invariance of direction oracles to strictly monotonically increasing (possibly nonaffine) transformations combined with the shortcoming of NBS and KSBS being sensitive to such nonaffine scalings. We provide an example below to help explain Proposition 1.

**Example 1.** Consider a bargaining game $\mathcal{B}_{[0,1]}([x^2, (x-1)^2], [1,1])$, for which both NBS and KSBS lie at $x = 1/2$. However, for the game with the nonaffine transformation $f(y) = y^2$ (monotonically strictly increasing in $\mathcal{S}$) applied to agent 1, $\mathcal{B}_{[0,1]}([x^4, (x-1)^2], [1,1])$ both NBS and KSBS are not at $x = 1/2$. However, in both games, the most preferred directions for agents 1 and 2 are to go towards $x = 0$, and $x = 1$ respectively. Thus, even if an algorithm employing only direction oracles finds NBS and KSBS for $\mathcal{B}_{[0,1]}([x^2, (x-1)^2], [1,1])$, it will not be able to find them for $\mathcal{B}_{[0,1]}([x^4, (x-1)^2], [1,1])$.

Thus, there is a clear need for introducing a new solution concept for bargaining problems which (i) can be identified by algorithms that employ only direction oracles, and (ii) is still robust to non-affine monotonic cost transformations.

## 2.2 Naive direction oracle-based bargaining can lead to unfair solutions

When the mediator has access to direction oracles, it is natural to construct a simple bargaining procedure which utilizes the sum of normalized gradients, i.e., at state $\mathbf{x}$, the mediator creates estimates of $\nabla \ell^i(\mathbf{x}) / \|\nabla \ell^i(\mathbf{x})\|_2, \forall i \in [N]$, and for some $\alpha > 0$, proceeds to the next state $\mathbf{x}^+$, given by

$$\mathbf{x}^+ = \mathbf{x} + \alpha \left( \sum_{i=1}^N \mathcal{O}_\ell^{\mathcal{D},i}(\mathbf{x}) \right) = \mathbf{x} - \alpha \left( \sum_{i=1}^N \frac{\nabla \ell^i(\mathbf{x})}{\|\nabla \ell^i(\mathbf{x})\|_2} \right). \tag{2}$$

At a glance, eq. (2) resembles a utilitarian approach to bargaining which would minimize the sum of agents' costs [26]; however, in fact the update rule in eq. (2) differs drastically as it weighs the direction associated with each agent $i$ equally. While eq. (2) corresponds to a valid bargaining solution satisfying Pareto stationarity (see Appendix D), we argue that this simplistic bargaining solution can lead to *unfair* solutions, as demonstrated by the following toy example.

**Example 2.** Consider again the two-agent bargaining game $\mathcal{B}_{[0,1]}([x^2, (x-1)^2], [1,1])$. Due to symmetry, a "fair" bargaining solution will reside at $x = 1/2$. Let eq. (2) be initialized at some $x_0 \in (0,1), x_0 \neq 1/2$. Then, because $\nabla \ell^1(x) / \|\nabla \ell^1(x)\|_2 = -\nabla \ell^2(x) / \|\nabla \ell^2(x)\|_2$, we get convergence at $x = x_0$, which is not a fair solution as $x_0$ can be initialized arbitrarily in $(0,1)$ far away from $x = 1/2$.

**Existing direction-oracle based bargaining algorithms give unfair solutions.** There are two existing works which attempt to conduct bargaining exclusively through directions, [7, 17]. Both works consider the two-agent setting, and propose iterative procedures for finding mutually beneficial directions, given the most preferred directions of both agents. However, both approaches consider bargaining scenarios with self-interested agents—which can lead to unfair solutions in cooperative bargaining settings; cf. Example 2, where agents never have a mutually beneficial direction. In such a situation, both algorithms stop wherever initialized and find points which are technically Pareto stationary, but heavily favor one agent. Further, [17] does not readily extend beyond the two-agent case in which finding mutually beneficial directions becomes combinatorially hard, and [7] reports a loss of mutual improvement and convergence guarantees in cases with more than two agents. A related but orthogonal line of work pertains to flocking in multi-agent scenarios [21], we discuss the differences between flocking and our bargaining solution in Appendix F.

## 3 Bargaining with Direction Oracles

It is clear that the mediator must take care when employing normalized gradients. The information which a solution concept like eq. (2) lacks is a notion of how far a potential solution is from each agent's preferred state. Existing bargaining solutions approach this issue by considering values in the *cost space* $\mathcal{L}$, which is not accessible in the direction oracle setting. Instead, in the direction oracle setting, one must conduct such reasoning in the *state space* $\mathcal{S}$. Even if the mediator has access to $\mathcal{L}$, reasoning in the state space can be beneficial as the components for each agent in $\mathcal{L}$ can be nonaffinely scaled, while $\mathcal{S}$ is shared uniformly by all agents.

A natural way to conduct such reasoning is to incorporate *how far* will the bargaining solution state be from each agent's preferred state (which is computable even in the direction oracle setting). To this end, we propose **Di**rection-based **B**argaining **S**olution (DiBS): starting from some state $\mathbf{x}_0$, DiBS conducts the following iterations to find a bargaining solution:

$$\mathbf{x}_{k+1} = \mathbf{f}(\mathbf{x}_k) := \mathbf{x}_k + \alpha_k \left( \sum_{i=1}^N \|\mathbf{x}_k - \mathbf{x}^{*,i}\|_2 \mathcal{O}_\ell^{\mathcal{D},i}(\mathbf{x}_k) \right)$$

$$= \mathbf{x}_k - \alpha_k \left( \sum_{i=1}^N \|\mathbf{x}_k - \mathbf{x}^{*,i}\|_2 \frac{\nabla \ell^i(\mathbf{x}_k)}{\|\nabla \ell^i(\mathbf{x}_k)\|_2} \right), \tag{DiBS}$$

where $\{\alpha_k\}_{k \geq 0}$ are appropriate step sizes, and $\mathbf{x}^{*,i} \in \arg\min_{\mathbf{x} \in \mathcal{S}} \ell^i(\mathbf{x})$ is a choice from the set of preferred states for the $i^{\text{th}}$ agent and fixed for all iterations. For the sake of convenience, if $\nabla \ell^i(\mathbf{x}) = 0$, we define $\nabla \ell^i(\mathbf{x}) / \|\nabla \ell^i(\mathbf{x})\|_2 = 0$. We remark that both the quantities used by DiBS, i.e., $\nabla \ell^i(\mathbf{x}) / \|\nabla \ell^i(\mathbf{x})\|_2$ and $\mathbf{x}^{*,i}$ are available through direction oracles which are implementable in practice

without using explicit agent costs—we elaborate upon this in Section 3.2. Note how `DiBS` differs from `NBS`: in its iterations, `NBS` gives more importance to those agents who have lower cost improvements (by scaling agent gradients with $1/d^i - \ell^i(\mathbf{x})$), while `DiBS` gives more importance to those agents who are further away from their preferred states (by scaling agent gradients with $\|\mathbf{x} - \mathbf{x}^{*,i}\|_2 / \|\nabla \ell^i(\mathbf{x})\|_2$).

Given the most preferred directions $\mathcal{O}_\ell^{\mathcal{D},i}(\mathbf{x}_k)$ and preferred states $\mathbf{x}^{*,i}$ of each agent, every iteration of `DiBS` has linear complexity in both the number of agents and the number of state dimensions. We remark that the most preferred states $\mathbf{x}^{*,i}$ can be efficiently calculated as a precomputation step by the mediator (if the agent is unable to directly provide them) using existing methods [13].

### 3.1 Theoretical properties of Direction-based Bargaining Solution (`DiBS`)

To establish the legitimacy of `DiBS` as a bargaining solution, we will first show that it finds Pareto stationary points. This can be seen by viewing `DiBS` as a dynamical system, and analyzing its fixed points. We define these concepts below. All proofs for our results can be found in Appendix A.

**Definition 3** (Fixed/Equilibrium Points)**.** Consider a function $\mathbf{f}_d : \mathbb{R}^n \to \mathbb{R}^n$ and the corresponding discrete-time dynamical system $\mathbf{x}_{k+1} = \mathbf{f}_d(\mathbf{x}_k)$. Then, $\tilde{\mathbf{x}}$ is a fixed point of the dynamical system $\mathbf{f}_d$ if $\mathbf{f}_d(\tilde{\mathbf{x}}) = \tilde{\mathbf{x}}$. Similarly, $\tilde{\mathbf{x}}$ is an equilibrium point of the continuous-time dynamical system $\dot{\mathbf{x}} = \mathbf{f}_c(\mathbf{x})$ for a function $\mathbf{f}_c : \mathbb{R}^n \to \mathbb{R}^n$ if $\mathbf{f}_c(\tilde{\mathbf{x}}) = 0$.

A dynamical system's ability to converge depends upon the system's *stability* properties. In particular, we will be interested in global asymptotic convergence.

**Definition 4** (Global Asymptotic Convergence)**.** Consider functions $\mathbf{f}_d : \mathbb{R}^n \to \mathbb{R}^n, \mathbf{f}_c : \mathbb{R}^n \to \mathbb{R}^n$ and a set $\mathcal{G} \subset \mathcal{S}$. The discrete-time dynamical system $\mathbf{x}_{k+1} = \mathbf{f}_d(\mathbf{x}_k), k \geq 0$ has global asymptotic convergence to $\mathcal{G}$ if $\lim_{k \to \infty} \mathbf{x}(k) \in \mathcal{G} \ \forall \ \mathbf{x}(0) \in \mathcal{S}$. Similarly, the continuous-time dynamical system $\dot{\mathbf{x}}(t) = \mathbf{f}_c(\mathbf{x}(t)), t \geq 0$ has global asymptotic convergence to $\mathcal{G}$ if $\lim_{t \to \infty} \mathbf{x}(t) \in \mathcal{G} \ \forall \ \mathbf{x}(0) \in \mathcal{S}$.

We begin by showing that any fixed point for `DiBS` is also Pareto stationary and that `DiBS` has global asymptotic convergence to the (non-empty) set of its fixed points (proof in Appendix A.2).

**Theorem 1** (Convergence of `DiBS` to Pareto Stationary Points)**.** *Direction-based Bargaining Solution (`DiBS`) has the following properties:*

1. *Any fixed point of `DiBS` is also a Pareto stationary point.*

2. *Under Assumption 2, the iterates of `DiBS` are bounded in $\mathbb{R}^n$.*

3. *Under Assumptions 1-2, a fixed point of `DiBS` always exists.*

4. *Assuming that agent costs $\ell^i$ are $\mu^i$-strongly convex and that Assumptions 1-2 hold, the continuous-time analog of `DiBS`,*

$$\dot{\mathbf{x}} = -h(\mathbf{x}) := -\sum_{i=1}^{N} \|\mathbf{x} - \mathbf{x}^{*,i}\|_2 \frac{\nabla \ell^i(\mathbf{x})}{\|\nabla \ell^i(\mathbf{x})\|_2}$$

*enjoys global asymptotic convergence to its equilibrium points. Further, if the agent costs are $L^i$-smooth, then `DiBS` enjoys global asymptotic convergence to the set of its (Pareto stationary) fixed points for stepsizes $\alpha_k > 0$ chosen such that $\sum_{k=0}^{\infty} \alpha_k = \infty$ and $\sum_{k=0}^{\infty} \alpha_k^2 < \infty$.*

Theorem 1 establishes that `DiBS` has desirable convergence properties, and is useful for finding Pareto stationary points when a mediator only utilizes direction oracles in a bargaining game. Now, we show that `DiBS` also inherits several key axioms from the cooperative bargaining literature (proof in Appendix A.3).

**Theorem 2** (Bargaining axioms satisfied by `DiBS`)**.** *The Direction-based Bargaining Solution (`DiBS`) satisfies the following axioms:*

1. *The solution found by `DiBS` is Pareto stationary (and strongly Pareto optimal if agent costs are twice differentiable and strictly convex).*

2. *The solution found by `DiBS` is invariant to strictly monotonically increasing nonaffine transformations.*

3. *The solution found by* `DiBS` *satisfies the Axiom of Symmetry (Axiom 2).*

4. *If* `DiBS` *has only one fixed point for a problem, then the solution found by* `DiBS` *satisfies the axiom of independence of irrelevant alternatives (Axiom 4).*

Importantly, Theorem 2 establishes the invariance of `DiBS` to monotonic *nonaffine* transformations, which is not obtained by `NBS` and `KSBS`. This invariance is particularly attractive as it allows `DiBS` to be robust against transformations which can occur due to imperfect cost function modeling or agent exaggerations, while still retaining the agents' relative preferences between states.

## 3.2 Practically obtaining a direction oracle

As mentioned earlier, it is possible to implement a direction oracle given only a binary *comparison* oracle, where at some state $\mathbf{x}$ in the bargaining game, the mediator asks every agent whether they prefer a different state $\mathbf{y}$ more than the current state, and the agents reply with the minimal information of "yes", "no", or "indifferent". Formally, a comparison oracle for the $i^{\text{th}}$ agent who is queried at a state $\mathbf{x}$ about a state $\mathbf{y}$, $\mathcal{O}_{\ell}^{\mathcal{C},i}(\mathbf{x}, \mathbf{y})$ is defined as

$$\mathcal{O}_{\ell}^{\mathcal{C},i}(\mathbf{x}, \mathbf{y}) = \begin{cases} +1, & \text{if } \ell^i(\mathbf{y}) < \ell^i(\mathbf{x}), \\ 0, & \text{if } \ell^i(\mathbf{y}) = \ell^i(\mathbf{x}), \\ -1, & \text{if } \ell^i(\mathbf{y}) > \ell^i(\mathbf{x}). \end{cases} \tag{3}$$

**What information can the comparison oracle give?** Optimization solely using comparison oracles is a well-studied problem in the single-agent setting. As a consequence, numerous algorithms exist in the literature which can estimate the normalized negative cost gradient, i.e., the most preferred direction, for an agent with only comparison oracles and can find the normalized gradients up to arbitrary accuracy for smooth functions. The number of queries required to estimate the normalized gradients up to a required accuracy is bounded, and many of the existing algorithms are also robust to noisy binary oracle evaluations [6, 10, 13, 24, 34]. This implies that if the $i^{\text{th}}$ agent's cost $\ell^i(\mathbf{x})$ is inaccessible to the mediator, the mediator can use any of the above off-the-shelf algorithms employing the minimal information comparison oracle as a practical way to estimate the agent's most preferred direction $-\nabla\ell^i(\mathbf{x})/\|\nabla\ell^i(\mathbf{x})\|_2$ with arbitrary accuracy. Further, the mediator can use the same algorithm to find the most preferred state $\mathbf{x}^{*,i} \in \arg\min_{\mathbf{x}\in\mathcal{S}} \ell^i(\mathbf{x})$ for the agent, details of which are given in Appendix E.

# 4 Experiments

We now evaluate our bargaining solution in practical problems. Our main aims are: (i) to investigate the solution quality of `DiBS`, (ii) to test the invariance of `DiBS` to monotonic nonaffine transformations, and (iii) to investigate the how the performance of `DiBS` is affected by the accuracy of normalized gradient estimates formed via comparison oracles.

## 4.1 Nonconvex multi-agent formation assignment under different bargaining solutions

In this experiment, $N$ agents, either `odd` or `even`, lie in a two-dimensional plane, and are attracted to a center point $\mathbf{c}$, while simultaneously exhibiting group-specific cohesion and repulsion behaviors. The position of the $i^{\text{th}}$ agent is $\mathbf{x}^i \in [0, 10] \times [0, 10] \subset \mathbb{R}^2$, and the game state is $\mathbf{x} = [\mathbf{x}^1, \ldots, \mathbf{x}^N]$, with $\mathcal{S} \subset \mathbb{R}^{2N}$. Agents with the same parity index (`odd` or `even`) prefer to remain close, while agents of different parities prefer greater separation. The $i^{\text{th}}$ agent's preferences are modeled using a cost function of the form

$$\ell^i(\mathbf{x}) = -a \cdot e^{-b\|\mathbf{x}^i - \mathbf{c}\|_2} - \sum_{j \neq i} \left( \left( e^{-\alpha^{ij}\|\mathbf{x}^i - \mathbf{x}^j\|_2} \right) - \left( e^{-\beta^{ij}\|\mathbf{x}^i - \mathbf{x}^j\|_2} \right) \right), a, b \in \mathbb{R}_+ \tag{4}$$

where the pairwise interaction weights $\alpha^{ij}, \beta^{ij} \in \mathbb{R}_+$ control group attraction and repulsion. We emphasize that the cost functions given in eq. (4) are nonconvex due to the difference of exponentials.

**Baselines.** We choose $N = 10$ agents, all implementation details can be found in Appendix C.1. We compare `DiBS` (with access to only direction oracles) against `NBS` and `KSBS` (both of which have

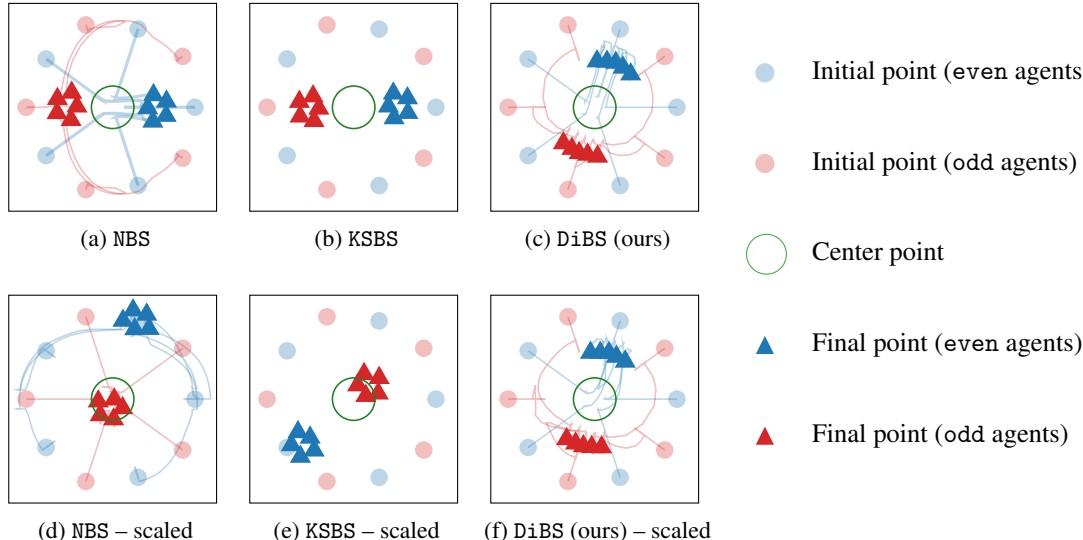

Figure 1: Formations achieved by different bargaining solutions. While DiBS yields qualitatively similar outcomes to NBS and KSBS in the original setting, it is also robust to monotone nonaffine scalings. KSBS is solved in a single shot, with no iteration trajectories to plot (see Appendix C.1)

access to the full costs $\ell^i$). We emphasize that DiBS, NBS and KSBS are different bargaining solution concepts, and while no concept is strictly "better" than the rest, we conduct this comparison to illustrate our motivation for developing DiBS.

DiBS **leads to balanced solutions.** Figure 1 (a)-(c) shows the initial and final positions of all agents. For DiBS and NBS, we also plot the variation of agent positions across iterations. KSBS is solved in one shot and does not have such variations available for plotting (see Appendix C.1). We observe that all three methods—NBS, KSBS, and DiBS reach reasonable solutions which respect agents' preferences, and balance their interests in different ways. At the solution, NBS and KSBS slightly prioritize clustering attractive agents while DiBS slightly prioritizes minimizing agent distances from the center, but overall all methods balance the three high-level objectives for all agents. Despite the nonconvex costs, we observe that all methods converge for the example.

**Invariance of DiBS to monotone nonaffine transformations.** Figure 1 (d)-(f) show the bargaining process when the costs given in eq. (4) for the odd agents undergo a monotone nonaffine transformation, i.e., $\text{sign}(l^i(\mathbf{x}))(l^i(\mathbf{x}))^2$, which retains the agents' relative preferences between states. As highlighted before, such transformations may occur due to a variety of reasons, such as modeling imperfections or exaggerated utilities. We observe that while NBS and KSBS completely change their solutions and present skewed, unfair outcomes that favor the odd agents with exaggerated utilities, our method DiBS still retains a fair outcome.

## 4.2 Mediated portfolio management through comparisons

In this experiment, we demonstrate the performance of DiBS where the direction oracle is approximated using comparisons.

**Setting.** A mediator allocates a shared stock investment fund across a set of $n$ stocks based on the preferences of a group of $N$ investors. The mediator's decision corresponds to a portfolio vector $\mathbf{x} \in \mathbb{R}^n$, where $\mathbf{x} \geq 0$ and $\mathbf{1}^\top \mathbf{x} = 1$. The $i^{\text{th}}$ investor has a cost function modeled using the well-known Markowitz portfolio theory [14], given by $\ell^i(\mathbf{x}) = \mathbf{x}^\top \Sigma^i \mathbf{x} - \lambda^i \mu^{i\top} \mathbf{x}$, where $\Sigma^i$ is the covariance matrix of stock returns, $\mu^i$ is the vector of expected returns, and $\lambda^i$ is the risk-reward tradeoff coefficient.

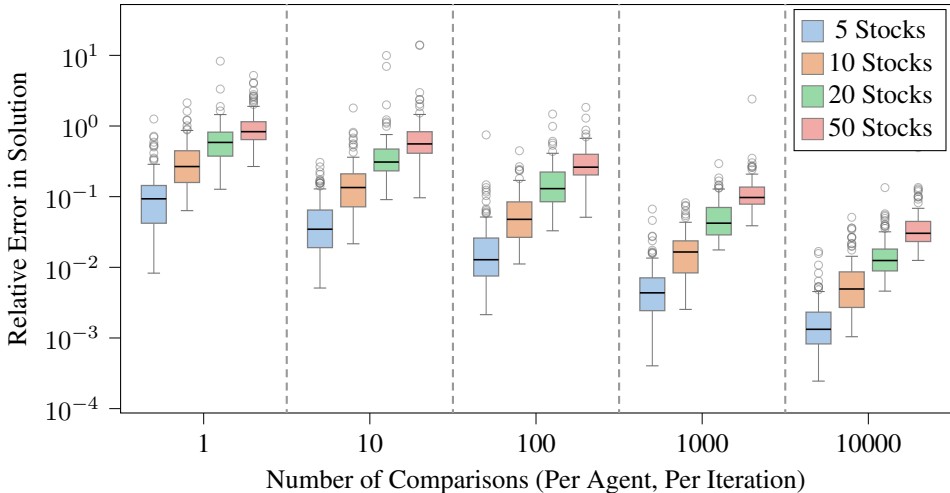

Figure 2: Results for the portfolio management example, showing the $1.5^{\text{th}}, 25^{\text{th}}, 50^{\text{th}}, 75^{\text{th}}$, and $98.5^{\text{th}}$ percentiles. DiBS offers promising performance even when the direction oracle is estimated through comparisons. Dots represent outliers; cf. Appendix C.2 for further details.

**Modeling diverse investor preferences.** The expected return vector $\mu^i$ and covariance matrix $\Sigma^i$ are computed from historical stock price data [2] in an investor-specific time window. Ultimately, each agent is assigned a personalized investment profile by randomly sampling:

- A time horizon from the following predefined investment windows: 5 days, 1 month, 3 months, 6 months, 1 year, 2 years, 5 years, or all time (8 years). All windows end on a common date of December 31, 2023.
- A risk-reward tradeoff coefficient $\lambda^i$ uniformly sampled from the interval $[0.0, \ 0.1]$.

This initialization results in agent-specific cost functions that reflect a diverse set of investor types and stock market views. We sample 100 scenarios in this manner. All implementation details are given in Appendix C.2. Additional results for different numbers of investors are in Appendix B.

**Baseline and metric.** We compare two versions of DiBS: one which uses the direction oracle yielding the solution $\mathbf{x}_{\text{dir}}^\dagger$, and one which uses comparisons to approximate direction oracles via the estimator used in Sign-OPT [6] yielding the solution $\mathbf{x}_{\text{comp}}^\dagger$. We allow this estimator to query the comparison oracle $1, 10, 100, 1000$ and $10000$ times at every iteration for every agent $i$. For both versions starting at $\mathbf{x}_0$, we calculate the relative error for each sample scenario, defined as $\frac{\|\mathbf{x}_{\text{dir}}^\dagger - \mathbf{x}_{\text{comp}}^\dagger\|}{\|\mathbf{x}_{\text{dir}}^\dagger - \mathbf{x}_0\|}$.

**DiBS offers promising performance even when the direction oracle is estimated through comparisons.** Figure 2 shows the relative errors for DiBS using comparisons vs. true direction oracles for $n = 5, 10, 20, 50$ stocks and $N = 10$ agents. We observe that, as expected, the accuracy increases with the number of comparisons allowed, and as the dimension of the problem (i.e., the number of stocks) increases, the number of comparisons which is required for accurate estimation of $\mathcal{O}_\ell^{\mathcal{D},i}$ increases. We remark that even when the number of comparisons allowed is significantly lower than the number of dimensions and, therefore, there is a significant error in the direction estimates, the median relative error of DiBS employing comparisons remains under 1, indicating improvement over the initial state towards the solution of DiBS employing exact directions.

## 5 Conclusion

We consider an increasingly important class of cooperative bargaining problems, in which mediators do not have access to agent utilities (which may be incomparable), and can instead only access agents' most preferred directions. These settings arise in human-AI interactions, privacy-sensitive

applications, and multi-agent interactions with exaggerated or imperfectly-modeled utilities. We show that no direction oracle-based algorithm can recover popular existing bargaining solutions (NBS, KSBS) for all bargaining games that satisfy standard assumptions. Therefore, we propose a new bargaining solution for this setting (DiBS), and show that it identifies Pareto stationary solutions, is invariant to monotonically increasing nonaffine transformations, and satisfies the axiom of symmetry. Under additional mild assumptions, we also show that DiBS satisfies the axiom of independence of irrelevant alternatives, and enjoys global asymptotic convergence to Pareto stationary solutions. Finally, we conduct experiments in two settings to validate our results and show that DiBS performs well when direction oracles are estimated using only comparison oracles, which are straightforward to implement in practice. Future work should investigate (i) relaxing the strong convexity assumption which is required in the proof of global convergence, (ii) providing non-asymptotic convergence results, and (iii) conducting experiments in large-scale learning settings.

## Acknowledgments and Disclosure of Funding

The authors would like to thank Filippos Fotiadis for insightful discussions on related topics. This research was supported by the Army Research Laboratory under cooperative agreements W911NF-23-2-0011 and W911NF-25-2-0021, the National Science Foundation under grant numbers 2336840 and 2211432, Office of Naval Research (ONR) grant ONR N00014-24-1-2797, and Army Research Office (ARO) grant ARO W911NF-23-1-0317. The views and conclusions contained in this document are those of the authors and should not be interpreted as representing the official policies, either expressed or implied, of these sponsors or the U.S. Government.

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

# Appendix table of contents

## A   Proofs

### A.1   Proof of Proposition 1

To prove Proposition 1, we first establish the invariance of the direction oracle presented in Equation (1) to strictly increasing monotonic (possibly nonaffine) transformations.

**Proposition 2.** *Consider an* $N-$*agent bargaining game* $\mathcal{B}_\mathcal{S}(\boldsymbol{\ell}, \mathbf{d})$ *with associated direction oracles* $\mathcal{O}_{\boldsymbol{\ell}}^{\mathcal{D},i}, i \in [N]$*. Let* $g^i : \mathbb{R} \to \mathbb{R}, i \in [N]$ *be strictly monotonically increasing, possibly nonaffine functions. Let* $\boldsymbol{g}(\boldsymbol{\ell})(\mathbf{x}) = [g^1(\ell^1(\mathbf{x})), \dots, g^N(\ell^N(\mathbf{x}))]$*. Then, for the direction oracles* $\mathcal{O}_{\boldsymbol{g}(\boldsymbol{\ell})}^{\mathcal{D},i}, i \in [N]$ *associated with the utility transformed bargaining game* $\mathcal{B}_\mathcal{S}(\boldsymbol{g}(\boldsymbol{\ell}), \mathbf{d})$*, we have* $\mathcal{O}_{\boldsymbol{\ell}}^{\mathcal{D},i} = \mathcal{O}_{\boldsymbol{g}(\boldsymbol{\ell})}^{\mathcal{D},i}, i \in [N]$*.*

*Proof.* For agent $j$, we have

$$\nabla g(\ell^j(\mathbf{x})) = \underbrace{g'(\ell^j(\mathbf{x}))}_{>0} \nabla \ell^j(\mathbf{x}) \qquad \text{(chain rule)}$$

$$\implies \frac{\nabla g(\ell^j(\mathbf{x}))}{\|\nabla g(\ell^j(\mathbf{x}))\|} = \frac{\ell^j(\mathbf{x})}{\|\ell^j(\mathbf{x})\|}$$

$$\implies \mathcal{O}_{\boldsymbol{g}(\boldsymbol{\ell})}^{\mathcal{D},i} = \mathcal{O}_{\boldsymbol{\ell}}^{\mathcal{D},i} \; \forall \; i \in [N]$$

$\square$

We can now prove proposition 1 by contradiction. Assume that there exists a deterministic algorithm $\mathcal{A}$ which can recover NBS or KSBS by only using direction oracles $\mathcal{O}_{\boldsymbol{\ell}}^{\mathcal{D},i}$ for all bargaining games $\mathcal{B}_\mathcal{S}(\boldsymbol{\ell}, \mathbf{d})$ satisfying Assumptions 1-2. Consider a nonaffine, strictly monotonically increasing function $g : \mathbb{R} \to \mathbb{R}$, with $g'(l) > 0 \; \forall \; l \in \mathbb{R}$, applied to transform only agent $j$'s utilities. Then, one can construct a new bargaining game $\mathcal{B}_\mathcal{S}(\tilde{\boldsymbol{\ell}}, \mathbf{d})$, where $\tilde{\boldsymbol{\ell}}$ corresponds to the costs

$$\tilde{\ell}^i(\mathbf{x}) = \begin{cases} \ell^i(\mathbf{x}), & i \neq j, \\ g(\ell^j(\mathbf{x})), & i = j. \end{cases}$$

As NBS and KSBS are not invariant to nonaffine transformations, one can choose $g$ such that NBS and KSBS for $\mathcal{B}_\mathcal{S}(\tilde{\boldsymbol{\ell}}, \mathbf{d})$ are different than the ones corresponding to $\mathcal{B}_\mathcal{S}(\boldsymbol{\ell}, \mathbf{d})$. However, from Proposition 2, we have $\mathcal{O}_{\tilde{\boldsymbol{\ell}}}^{\mathcal{D},i} = \mathcal{O}_{\boldsymbol{\ell}}^{\mathcal{D},i} \; \forall \; i \in [N]$. When algorithm $\mathcal{A}$ is used to solve the bargaining problem $\mathcal{B}_\mathcal{S}(\tilde{\boldsymbol{\ell}}, \mathbf{d})$, $\mathcal{A}$ can query $\mathcal{O}_{\tilde{\boldsymbol{\ell}}}^{\mathcal{D},i}, i \in [N]$. However, because $\mathcal{O}_{\tilde{\boldsymbol{\ell}}}^{\mathcal{D},i} = \mathcal{O}_{\boldsymbol{\ell}}^{\mathcal{D},i}$ and $\mathcal{A}$ is a deterministic algorithm, $\mathcal{A}$ will return the same solution as it did for $\mathcal{B}_\mathcal{S}(\boldsymbol{\ell}, \mathbf{d})$, which cannot be the bargaining solution for $\mathcal{B}_\mathcal{S}(\tilde{\boldsymbol{\ell}}, \mathbf{d})$ because of the nonaffine transformation $g$. Hence, by contradiction, there is no such $\mathcal{A}$.

## A.2 Proof of Theorem 1

**Proof Sketch:** We first show that the iterates of `DiBS` are bounded, and that a fixed point of `DiBS` always exists. We then show that any fixed point of `DiBS` is also Pareto stationary. Then we proceed to analyze the continuous-time dynamical system corresponding to `DiBS`, show that it enjoys global asymptotic converge to it's equilbrium points (fixed points of `DiBS`). Finally, we show that there is a way to choose step size such that `DiBS` retains these convergence properties.

1. **Iterates of `DiBS` remain bounded.** Consider a ball $\mathcal{B}$ in $\mathcal{S}$ with finite radius big enough to contain $\mathbf{x}^{*,i} \ \forall \ i \in [N]$. If `DiBS` iterates diverge, they must escape this ball. Consider the situation when the `DiBS` iterates are at the boundary of this ball at some point $\mathbf{x}$. Consider the vectors

$$g^i = \frac{\nabla \ell^i(\mathbf{x})}{\|\nabla \ell^i(\mathbf{x})\|_2} \|\mathbf{x} - \mathbf{x}^{*,i}\|_2.$$

   Because $\mathbf{x}^{*,i} \in \mathcal{B}, i \in [N]$, all $g^i, i = 1, \dots, N$ point inside the ball $\mathcal{B}$. Then the quantity $\sum_{i=1}^{N} g^i$ lies in the convex cone of $g^i$'s and must also point inwards the ball $\mathcal{B}$. Thus the next `DiBS` iterate must lie within the ball $\mathcal{B}$, can never escape this ball of finite radius and remain in $\mathcal{S}$.

2. **A fixed point of `DiBS` always exists.** Using the fact that the `DiBS` iterates remain bounded in $\mathcal{S}$, and the convexity of the Euclidean ball, from Brouwer's fixed point theorem [5], we get that a fixed point of `DiBS` always exists in $\mathcal{S}$.

3. **Fixed points of `DiBS` are Pareto stationary.** Let $\mathbf{x}^\dagger$ be a fixed point of `DiBS`, then for some $\alpha > 0$,

$$\mathbf{x}^\dagger = \mathbf{x}^\dagger - \alpha \sum_{i=1}^{N} \frac{\nabla \ell^i(\mathbf{x}^\dagger)}{\|\nabla \ell^i(\mathbf{x}^\dagger)\|_2} \|\mathbf{x}^\dagger - \mathbf{x}^{*,i}\|_2$$

$$\implies \sum_{i=1}^{N} \frac{\nabla \ell^i(\mathbf{x}^\dagger)}{\|\nabla \ell^i(\mathbf{x}^\dagger)\|_2} \|\mathbf{x}^\dagger - \mathbf{x}^{*,i}\|_2 = 0,$$

   which satisfies the definition of Pareto stationarity given in Definition 1 with

$$\beta^i = \frac{\|\mathbf{x}^\dagger - \mathbf{x}^{*,i}\|_2 / \|\nabla \ell^i(\mathbf{x}^\dagger)\|_2}{\sum_{i=1}^{N} \|\mathbf{x}^\dagger - \mathbf{x}^{*,i}\|_2 / \|\nabla \ell^i(\mathbf{x}^\dagger)\|_2}.$$

4. **Global asymptotic convergence of continuous-time analog of `DiBS`.** Consider the continuous time dynamics corresponding to `DiBS`, given by

$$\dot{\mathbf{x}} = -h(\mathbf{x}) := -\sum_{i=1}^{N} \frac{\nabla \ell^i(\mathbf{x})}{\|\nabla \ell^i(\mathbf{x})\|_2} \|\mathbf{x} - \mathbf{x}^{*,i}\|_2.$$

   At an equilibrium point $\mathbf{x}^\dagger$ of $h$, we have that $h(\mathbf{x}^\dagger) = 0$. Now consider for the $i^{\text{th}}$ agent,

$$h_i(\mathbf{x}) := \frac{\nabla \ell^i(\mathbf{x})}{\|\nabla \ell^i(\mathbf{x})\|_2} \|\mathbf{x} - \mathbf{x}^{*,i}\|_2$$

$$\implies \nabla h_i(\mathbf{x}) = \nabla \ell^i(\mathbf{x}) \left( \nabla \left( \frac{\|\mathbf{x} - \mathbf{x}^{*,i}\|_2}{\|\nabla \ell^i(\mathbf{x})\|_2} \right) \right)^\top + \frac{\|\mathbf{x} - \mathbf{x}^{*,i}\|_2}{\|\nabla \ell^i(\mathbf{x})\|_2} H_i(\mathbf{x}), \ H_i(\mathbf{x}) := \nabla^2 \ell^i(\mathbf{x})$$

$$= \underbrace{\frac{\nabla \ell^i(\mathbf{x})(\mathbf{x} - \mathbf{x}^{*,i})^\top}{\|\nabla \ell^i(\mathbf{x})\|_2 \|\mathbf{x} - \mathbf{x}^{*,i}\|_2}}_{:=A(\mathbf{x})} + \underbrace{\frac{\|\mathbf{x} - \mathbf{x}^{*,i}\|_2}{\|\nabla \ell^i(\mathbf{x})\|_2} \left( I - \frac{\nabla \ell^i(\mathbf{x}) \nabla \ell^i(\mathbf{x})^\top}{\|\nabla \ell^i(\mathbf{x})\|_2^2} \right) H_i(\mathbf{x})}_{:=B(\mathbf{x})}$$

   We will show that for all agents $i$, $\mathbf{u}^\top \left( \nabla h_i(\mathbf{x}) + \nabla h_i(\mathbf{x})^\top \right) \mathbf{u} < 0 \ \forall \ \mathbf{u}$ such that $h(\mathbf{u}) \neq 0, \mathbf{u} \in \mathbb{R}^n$. For any $\mathbf{u} \parallel \nabla \ell^i(\mathbf{x})$, we have $\mathbf{u}^\top B(\mathbf{x}) = 0 \implies \mathbf{u}^\top B(\mathbf{x})\mathbf{u} = 0$. For any

$\mathbf{u} \perp \nabla\ell^i(\mathbf{x})$, we have

$$\mathbf{u}^\top B(\mathbf{x})\mathbf{u} = \frac{\|\mathbf{x} - \mathbf{x}^{*,i}\|_2}{\|\nabla\ell^i(\mathbf{x})\|_2}\mathbf{u}^\top \left(I - \frac{\nabla\ell^i(\mathbf{x})\nabla\ell^i(\mathbf{x})^\top}{\|\nabla\ell^i(\mathbf{x})\|_2^2}\right) H_i(\mathbf{x})\mathbf{u}$$

$$= \frac{\|\mathbf{x} - \mathbf{x}^{*,i}\|_2}{\|\nabla\ell^i(\mathbf{x})\|_2}\mathbf{u}^\top \left(I - \frac{\nabla\ell^i(\mathbf{x})\nabla\ell^i(\mathbf{x})^\top}{\|\nabla\ell^i(\mathbf{x})\|_2^2}\right) H_i(\mathbf{x}) \left(I - \frac{\nabla\ell^i(\mathbf{x})\nabla\ell^i(\mathbf{x})^\top}{\|\nabla\ell^i(\mathbf{x})\|_2^2}\right)\mathbf{u}$$

$$\quad\quad\quad\quad\quad\quad\quad\quad\quad\quad\quad\quad\quad\quad\quad\quad\quad\quad\quad\quad\quad (\mathbf{u} \perp \nabla\ell^i(\mathbf{x}))$$

$$> 0 \,\forall\, \mathbf{x} \neq \mathbf{x}^{*,i}, \quad\quad\quad\quad\quad\quad\quad\quad\quad\quad\quad\quad (H_i \succ \mu^i I)$$

$$\implies \mathbf{u}^\top B(\mathbf{x})\mathbf{u} = \begin{cases} 0, & \mathbf{u} \parallel \nabla\ell^i(\mathbf{x}) \\ > 0 \,\forall\, \mathbf{x} \neq \mathbf{x}^{*,i}, & \mathbf{u} \perp \nabla\ell^i(\mathbf{x}) \end{cases} \quad\quad\quad (5)$$

with a similar line of logic for $\mathbf{u}^\top B(\mathbf{x})^\top\mathbf{u}$. Now we will consider $A(\mathbf{x})$. We have, for $\mathbf{u} \perp \nabla\ell^i(\mathbf{x})$, $\mathbf{u}^\top A(\mathbf{x}) = 0 \implies \mathbf{u}^\top A\mathbf{u} = 0$. For $\mathbf{u} \parallel \nabla\ell^i(\mathbf{x})$, let $\mathbf{u} = \gamma\ell^i(\mathbf{x}), \gamma \in \mathbb{R}\backslash\{0\}$, we have

$$\mathbf{u}^\top A(\mathbf{x})\mathbf{u} = \gamma^2\|\nabla\ell^i(\mathbf{x})\|(\mathbf{x} - \mathbf{x}^{*,i})^\top\nabla\ell^i(\mathbf{x}) > 0 \,\forall\, \mathbf{x} \neq \mathbf{x}^{*,i}.$$

$$\text{(from strong convexity assumption)}$$

$$\implies \mathbf{u}^\top A(\mathbf{x})\mathbf{u} = \begin{cases} > 0 \,\forall\, \mathbf{x} \neq \mathbf{x}^{*,i}, & \mathbf{u} \parallel \nabla\ell^i(\mathbf{x}) \\ 0, & \mathbf{u} \perp \nabla\ell^i(\mathbf{x}), \end{cases} \quad\quad\quad (6)$$

with similar logic for $\mathbf{u}^\top A(\mathbf{x})^\top\mathbf{u}$. Combining eq. (5) and eq. (6), we get that

$$-\mathbf{u}^\top \left(\nabla h_i(\mathbf{x}) + \nabla h_i(\mathbf{x})^\top\right)\mathbf{u} > 0 \,\forall\, \mathbf{u} \in \mathbb{R}^n \setminus \{0, \mathbf{x}^{*,i}\}$$

$$\implies \mathbf{u}^\top \left(\nabla h(\mathbf{x}) + \nabla h(\mathbf{x})^\top\right)\mathbf{u} = -\sum_{i=1}^N \mathbf{u}^\top \left(\nabla h_i(\mathbf{x}) + \nabla h_i(\mathbf{x})^\top\right)\mathbf{u} < 0 \,\forall\, \mathbf{u} \in \mathbb{R}^n \setminus \{0\}$$

$$(7)$$

Now, let us make a Lyapunov function $V(\mathbf{x}) : \mathbb{R}^n \to \mathbb{R}$ for $h(\mathbf{x})$ given by

$$V(\mathbf{x}) = h(\mathbf{x})^\top h(\mathbf{x})$$

$$\implies \dot{V}(\mathbf{x}) = h(\mathbf{x})^\top \left(\nabla h(\mathbf{x}) + \nabla h(\mathbf{x})^\top\right) h(\mathbf{x})$$

Further, we have $V(\mathbf{x}) \to \infty$ as $\|\mathbf{x}\| \to \infty$. Further from eq. (7), $\dot{V}(\mathbf{x}) \leq 0$, with $\dot{V}(\mathbf{x}) = 0$ only when $\mathbf{x}$ is an equilibrium of $h$. Thus, by classical results in nonlinear systems theory, we have that all equilibrium points of $\dot{x} = -h(\mathbf{x})$ are local asymptotically stable, and by LaSalle's invariance theorem we have that the system the continuous time dynamics $\dot{\mathbf{x}} = -h(\mathbf{x})$ converges globally asymptotically to the set of equilibrium points [12, Theorem 4.4].

5. DiBS **retains continuous-time guarantees with correct step sizes.** We have that the mapping $h(\mathbf{x}) : \mathbb{R}^n \to \mathbb{R}$ is $L_h$−Lipschitz for some $L_h > 0$. Then, using any square summable sequence of step sizes $\alpha_k > 0$ such that $\sum_{k=0}^\infty \alpha_k = \infty, \sum_{k=0}^\infty \alpha_k^2 < \infty$ retains the global convergence properties for DiBS [4, Chapter 2]. To see the Lipschitzness of $h$, we have

$$h_i(\mathbf{x}) - h_i(\mathbf{y})$$

$$= \left\|\frac{\nabla\ell^i(\mathbf{x})}{\|\nabla\ell^i(\mathbf{x})\|_2}\|\mathbf{x} - \mathbf{x}^{*,i}\|_2 - \frac{\nabla\ell^i(\mathbf{y})}{\|\nabla\ell^i(\mathbf{y})\|_2}\|\mathbf{y} - \mathbf{x}^{*,i}\|_2\right\|_2$$

$$= \left\|\left(\frac{\nabla\ell^i(\mathbf{x})}{\|\nabla\ell^i(\mathbf{x})\|_2} - \frac{\nabla\ell^i(\mathbf{y})}{\|\nabla\ell^i(\mathbf{y})\|_2}\right)\|\mathbf{x} - \mathbf{x}^{*,i}\|_2 - \frac{\nabla\ell^i(\mathbf{y})}{\|\nabla\ell^i(\mathbf{y})\|_2}\left(\|\mathbf{y} - \mathbf{x}^{*,i}\|_2 - \|\mathbf{x} - \mathbf{x}^{*,i}\|_2\right)\right\|_2$$

$$\leq \underbrace{\left\|\left(\frac{\nabla\ell^i(\mathbf{x})}{\|\nabla\ell^i(\mathbf{x})\|_2} - \frac{\nabla\ell^i(\mathbf{y})}{\|\nabla\ell^i(\mathbf{y})\|_2}\right)\|\mathbf{x} - \mathbf{x}^{*,i}\|_2\right\|_2}_{:=\text{I}} + \underbrace{\left\|\frac{\nabla\ell^i(\mathbf{y})}{\|\nabla\ell^i(\mathbf{y})\|_2}\left(\|\mathbf{y} - \mathbf{x}^{*,i}\|_2 - \|\mathbf{x} - \mathbf{x}^{*,i}\|_2\right)\right\|_2}_{:=\text{II}}$$

Bounding term II, we have

$$\text{II} \leq \left\| \frac{\nabla \ell^i(\mathbf{y})}{\|\nabla \ell^i(\mathbf{y})\|_2} \right\|_2 \|\mathbf{x} - \mathbf{y}\|_2 = \|\mathbf{x} - \mathbf{y}\|_2 \qquad \text{(reverse triangle inequality)}$$

Now for term I, we have

$$\left\| \frac{\nabla \ell^i(\mathbf{x})}{\|\nabla \ell^i(\mathbf{x})\|_2} - \frac{\nabla \ell^i(\mathbf{y})}{\|\nabla \ell^i(\mathbf{y})\|_2} \right\| = \left\| \frac{\|\nabla \ell^i(\mathbf{y})\|_2 \nabla \ell^i(\mathbf{x}) - \|\nabla \ell^i(\mathbf{x})\|_2 \nabla \ell^i(\mathbf{y})}{\|\nabla \ell^i(\mathbf{x})\|_2 \|\nabla \ell^i(\mathbf{y})\|_2} \right\|$$

$$= \left\| \frac{\|\nabla \ell^i(\mathbf{y})\|_2 \left( \nabla \ell^i(\mathbf{x}) - \nabla \ell^i(\mathbf{y}) \right) + \left( \|\nabla \ell^i(\mathbf{y})\|_2 - \|\nabla \ell^i(\mathbf{x})\|_2 \right) \nabla \ell^i(\mathbf{y})}{\|\nabla \ell^i(\mathbf{x})\|_2 \|\nabla \ell^i(\mathbf{y})\|_2} \right\|$$

$$\leq 2 \frac{\|\nabla \ell^i(\mathbf{y})\|_2 \|\nabla \ell^i(\mathbf{x}) - \nabla \ell^i(\mathbf{y})\|_2}{\|\nabla \ell^i(\mathbf{x})\|_2 \|\nabla \ell^i(\mathbf{y})\|_2} = 2 \frac{\|\nabla \ell^i(\mathbf{x}) - \nabla \ell^i(\mathbf{y})\|_2}{\|\nabla \ell^i(\mathbf{x})\|_2}$$

Now from $\mu^i-$strong convexity of $\ell^i(\cdot)$,

$$\|\nabla \ell^i(\mathbf{x})\| \geq \mu^i \|\mathbf{x} - \mathbf{x}^{*,i}\|$$

$$\implies \frac{1}{\|\nabla \ell^i(\mathbf{x})\|_2} \leq \frac{1}{\mu^i \|\mathbf{x} - \mathbf{x}^{*,i}\|_2}$$

Further, from $L^i-$smoothness of $\ell^i(\cdot)$, we have $\|\nabla \ell^i(\mathbf{x}) - \nabla \ell^i(\mathbf{y})\|_2 \leq L^i \|\mathbf{x} - \mathbf{y}\|$. Thus, we have

$$\text{I} \leq \frac{2L^i}{\mu^i} \|\mathbf{x} - \mathbf{y}\|_2$$

Combining these bounds for I and II, we get

$$\|h_i(\mathbf{x}) - h_i(\mathbf{y})\|_2 \leq \left( \frac{2L^i}{\mu^i} + 1 \right) \|\mathbf{x} - \mathbf{y}\|_2$$

Summing over all agents, we have

$$L_h = \sum_{i=1}^{N} \left( 1 + \frac{2L^i}{\mu^i} \right)$$

Thus, $h$ is $L_h-$Lipschitz.

## A.3 Proof of Theorem 2

1. **Pareto.** The fact that `DiBS` finds Pareto stationary solutions follows directly from Theorem 1. Pareto stationarity is a necessary condition for Pareto optimality, and a sufficient condition when agent costs are strictly convex and twice differentiable [15, 23].

2. **Invariance.** For invariance to strictly increasing monotonic functions, let a nonaffine transformation $g^i : \mathbb{R} \to \mathbb{R}, g'(l) > 0 \; \forall \; l \in \mathbb{R}$ be applied to $\ell^i(\mathbf{x})$. Let $\tilde{\ell}(\mathbf{x}) = [g^1(\ell^1), \ldots, g^N(\ell^N)]$. Then as in the proof of Proposition 1, we have

$$\nabla \tilde{\ell^i}(\mathbf{x}) = \nabla g(\ell^i(\mathbf{x})) = \underbrace{g'(\ell^i(\mathbf{x}))}_{>0} \nabla \ell^i(\mathbf{x})$$

$$\implies \frac{\nabla \tilde{\ell^i}(\mathbf{x})}{\|\nabla \tilde{\ell^i}(\mathbf{x})\|_2} = \frac{\nabla \ell^i(\mathbf{x})}{\|\nabla \ell^i(\mathbf{x})\|_2}$$

$$\implies \mathcal{O}_{\tilde{\ell}}^{\mathcal{D},i} = \mathcal{O}_{\ell}^{\mathcal{D},i} \; \forall \; i \in [N].$$

Further, because of monotonicity, $\arg\min_{\mathbf{x}} g(\ell^i(\mathbf{x})) = \arg\min_{\mathbf{x}} \ell^i(\mathbf{x}) = \mathbf{x}^{*,i}$. Thus, both the pieces of information used for each player by `DiBS` remains invariant to the transformation for each agent. This leads to invariance of `DiBS` against strictly monotonic nonaffine transformations.

3. **Symmetry.** It is trivial to see that `DiBS` satisfies the axiom of symmetry because the `DiBS` takes the input from each agent at the same state during each iteration. It is invariant to permutations of agents.

4. **Independence of Irrelevant Alternatives.** We know that `DiBS` is globally asymptotically convergent to the set of its fixed points from Theorem 1. This means that if there is only a single fixed point, `DiBS` will have global asymptotic convergence to this point, which satisfies Axiom 4.

# B   Additional experiments for mediated portfolio management

We include the results for the mediated portfolio management experiment repeated for $N = 2, 3$ and $5$ investor agents. These experiments have similar trends as mentioned in Section 4.2. The result plots are attached here in Figures 3, 4, 5.

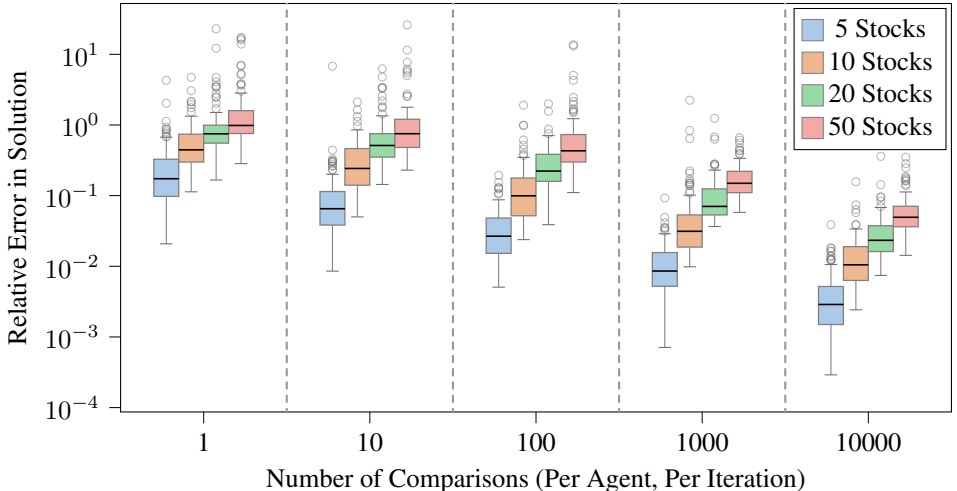

Figure 3: Repeating the Mediated Portfolio Management experiment for $N = 2$ agents.

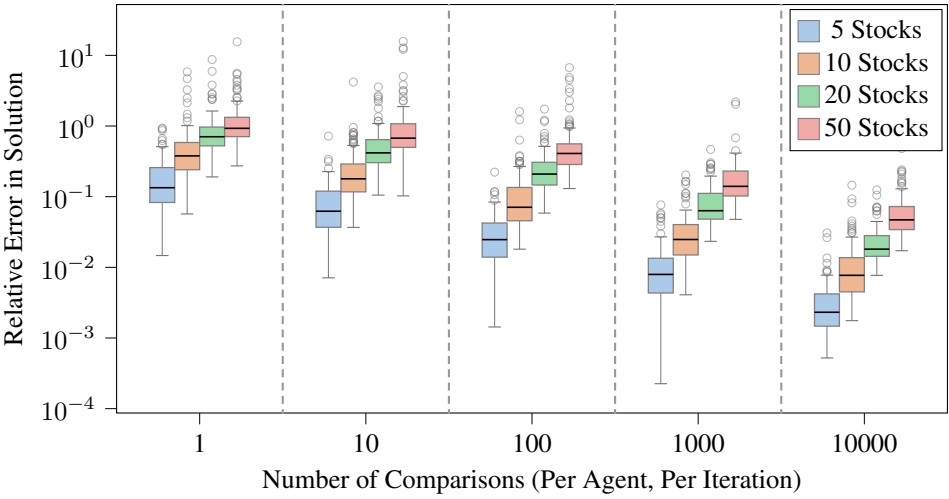

Figure 4: Repeating the Mediated Portfolio Management experiment for $N = 3$ agents.

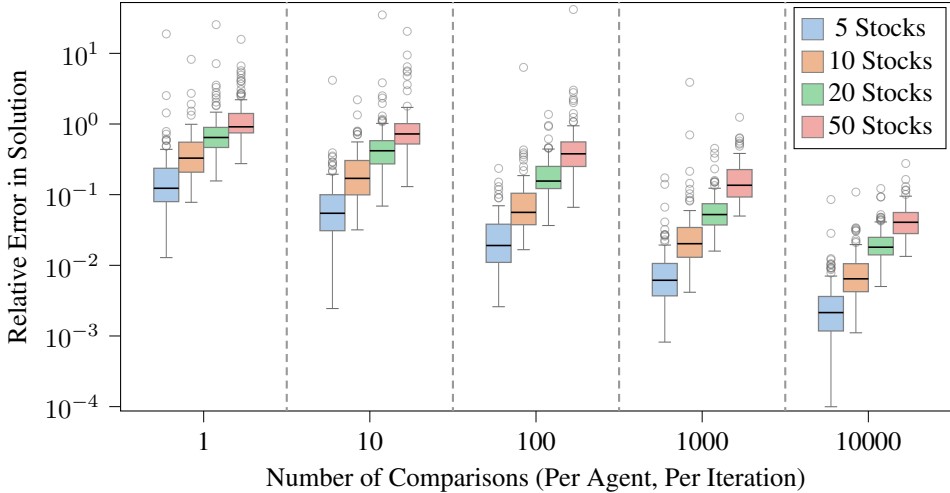

Figure 5: Repeating the Mediated Portfolio Management experiment for $N = 5$ agents.

## C  Experimental details

### C.1  Multi-agent formation assignment implementation details

**Parameter values.**  We choose $\mathbf{c} = (5, 5)$, $a = 10$ and $b = 0.01$. The agents were initialized in a circle centered at $\mathbf{c}$, with a radius of $3$. The value of the group attraction and repulsion values used for the experiment are

$$\alpha_{ij} = \begin{cases} 1 & \text{if } i \text{ and } j \text{ are both odd or even} \\ 0.1 & \text{otherwise} \end{cases}$$

$$\beta_{ij} = \begin{cases} 3 & \text{if } i \text{ and } j \text{ are both odd or even} \\ 0.9 & \text{otherwise.} \end{cases}$$

Based on these values, agents want to maintain a distance of $0.5493$ with agents of the same group and a distance of $2.7465$ with agents of the other group.

**Algorithmic Details.**  `DiBS` and `NBS` were both run for 5000 iterations. `KSBS` was solved in one shot, as it is based on a geometric argument with no iterative scheme. To solve `KSBS`, we minimized the sum of loss improvements $\gamma^i$ for each agent $i$, while ensuring equal loss improvements for all agents. This was encoded in the objective

$$\mathcal{L}(\boldsymbol{\gamma}) = \sum_{i=1}^{N} \gamma^i - \left\| \boldsymbol{\gamma} - \frac{1}{N} \sum_{j=1}^{N} \gamma^j \cdot \mathbf{1} \right\|_2, \boldsymbol{\gamma} = [\gamma^1, \dots, \gamma^N].$$

For `NBS` and `KSBS`, $d^i = 0 \ \forall \ i \in [N]$.

### C.2  Mediated portfolio management implementation details

**Implementation details.**  We conducted 100 random initializations for each number of stocks $(5, 10, 20, 50)$. Every random initialization was run for $1, 10, 100, 1000, 10000$ comparisons made per agent per iteration. Real life stock data was procured sing the `yfinance` Python package [2] (under the Apache license). We ensured that the simplex constraints for this example were met by using the following strategies:

1. Projecting all agent gradients onto the simplex before performing an update.
2. Shrinking the step size by a factor of $10$ if a step would cause any element in the state to become less than zero. If the step size becomes less than $10^{-12}$, we stop the algorithm. The initial step size was set to be $0.01$.

For terminating the algorithms, we used the termination condition of either the step size reaching $10^{-12}$, or the algorithm completing 1000 iterations.

As mentioned, the box plot was made using 100 random initializations for each number of stocks $(5, 10, 20, 50)$. In the box plots for Figures 2, 3, 4 and 5, outliers (dots) were chosen to be data points that were below $Q_1 - 1.5(Q_3 - Q_1)$ or above $Q_3 + 1.5(Q_3 - Q_1)$. Here, $Q_1, Q_3$ denote the first and third quartiles respectively.

### C.3 Hardware Details

All experiments were run on a desktop with a 12th Gen Intel(R) Core(TM) i7-12700 12-core CPU.

## D  On naive bargaining algorithm given in Equation 2

The solution found by the iterates of the naive bargaining algorithm given in eq. (2) satisfy

1. Pareto Stationarity: This is because if its iterates converge at some point $\mathbf{x}$, we have for eq. (2) that $\sum_i \frac{\nabla \ell^i(\mathbf{x})}{\|\nabla \ell^i(\mathbf{x})\|_2} = 0$, which satisfies Definition 1 with $\beta_i = 1/N$.

2. Symmetry: This is trivial to see because eq. (2) is invariant to permuting the agents' order.

3. Invariance to monotone nonaffine transformations: this follows for eq. (2) from the proof of Proposition 1.

## E  Obtaining preferred states using direction oracles

In this section, we include a more in-depth discussion on how one can obtain preferred states using only direction oracles. Recall that each agent $i$ provides access to a *direction oracle* $\mathcal{O}_{\boldsymbol{\ell}}^{\mathcal{D},i}(\mathbf{x}) = -\frac{\nabla \ell^i(\mathbf{x})}{\|\nabla \ell^i(\mathbf{x})\|_2}$ that specify the direction of steepest descent for an agent's objective $\ell^i$. Despite not observing $\ell^i$ or $\nabla \ell^i$ directly, several results from the zeroth- and first-order optimization literature show that preferred (locally optimal) states can be recovered using only such directional feedback.

A simple and widely studied approach is to perform a gradient descent-like update of the form:

$$\mathbf{x}_{t+1} = \mathbf{x}_t + \eta \mathcal{O}_{\boldsymbol{\ell}}^{\mathcal{D},i}(\mathbf{x}_t),$$

where $\eta$ is a suitably chosen step size. Methods that use this update, including SIGNSGD [3] and SIGN-OPT [6], have been shown to converge to stationary points for smooth functions. Under Lipschitz and smoothness assumptions, these methods satisfy bounds of the form

$$\mathbb{E}\big[\|\nabla \ell^i(\mathbf{x}_T)\|^2\big] = \mathcal{O}\left(\frac{\sqrt{n}}{\sqrt{T}} + \frac{n}{\sqrt{Q}}\right),$$

where $n$ is the dimension of state $\mathbf{x}$, $T$ is the number of descent iterations, and $Q$ the number of sampled directional queries per iteration.

## F  Relation to multi-agent consensus and flocking.

At a structural level, the DiBS update may appear reminiscent of distributed consensus or flocking dynamics [21], which also involve averaging normalized direction vectors across agents. However, these methods assume specific inter-agent potential functions and neighborhood graphs that govern attraction and alignment behaviors. In contrast, DiBS does not assume any specific potential function for the agents. Each agent possesses an independent cost function $\ell_i(x)$, and the mediator aggregates their direction oracles using dynamic weights that depend on distances to their respective preferred states. Consequently, DiBS generalizes beyond consensus-seeking to a general cooperative bargaining framework that aims to achieve Pareto-stationary and fair outcomes rather than spatial alignment or agreement.

