# OpenReview forum: "Cooperative Bargaining Games Without Utilities: Mediated Solutions from Direction Oracles"
_NeurIPS.cc/2025/Conference — NeurIPS 2025 poster_

### Official Review · Reviewer_XLAd · 2025-06-08

**Clarity:** 3
**Significance:** 3
**Originality:** 3
**Rating:** 5
**Confidence:** 4

**Summary:**

The paper presents a cooperative bargaining solution satisfying Pareto stationarity, invariance to monotonic utility transformations, symmetry, and IIA. This solution concept can be found with a new protocol, DiBS, which has access to only utility gradients for each agent. This expands beyond traditional approaches, NBS and KSBS, which require more information from the agents in order to compute. Direction oracles can be found practically using comparison queries. Experiments demonstrate the validity of DiBS and some failures by the NBS and KSBS solutions when utilities are scaled in a non-affine manner.

**Questions:**

- On line 274 - can you please clarify or cite a specific query complexity for estimating the normalized gradient?
- Can you comment on how theoretically accurate a direction oracle approach would be for computing DiBS? This question appears to be what is experimentally validated through Sec 4.2.
- In Sec 4.2, can you please clarify how this example is a cooperative bargaining game? Whereas it was clear how the example in Sec 4.1 compared DiBS to NBS and KSBS, I'm not immediately seeing how you're computing DiBS, given the error on line 337.
- (Extensions) As you mention on line 269, "Optimization solely using comparison oracles is a well-studied problem in the single-agent setting." What you suggest in this paper seems like an application of comparison/direction oracles into the multiagent space. I'm wondering what other types of problems you think this technique could be applied to beyond cooperative bargaining.

**Ethical Concerns:**

["NO or VERY MINOR ethics concerns only"]

**Final Justification:**

The rebuttal sufficiently addressed my comments. I will keep my score positively as-is.

**Quality:**

3

**Strengths And Weaknesses:**

Strengths:

Overall the paper presents an interesting formulation of the bargaining game and noteworthy extension, where the mechanism only has access to gradients as opposed to full utility information from the agents. The paper presents the problem coherently, demonstrates two theorems satisfied by DiBS, and how to feasibly obtain gradient information from agents. I am not left with any significant questions after reading this work.


Weaknesses:

My main concern of the paper is only if DiBS (line 207) has already introduced in the multiagent control systems literature. I can't point to a specific paper, but I would think that the solution {normalized gradient weighted by individual agent loss} would have already been proposed, perhaps pertaining to the problem of distributed averaging (related to Flocking by Olfati-Saber, 2006). Are you familiar with this span of literature?

Also, global asymptotic convergence may not be the most surprising finding  given that the loss functions are assumed to be smooth (Thm 1.4) and convex (Assumption 2). This is noted by [3, Chapter 2] on line 491. But overall this paper does seem like a worthwhile contribution to the literature.


Minor:
- Line 114: Please explain or cite why these iterates converge to NBS.
- Please elaborate on how SignSGD can be used to compute the most preferred states, on Line 218, for readers who are not familiar.
- Why is proof of Theorem 1 a proof "sketch" (line 452)?

---

> ### Author Rebuttal · Authors · 2025-07-31
>
> Thank you for your review and comments! Please find our responses below.
>
> ``My main concern of the paper is only if DiBS (line 207) has already introduced in the multiagent control systems literature ...... perhaps pertaining to the problem of distributed averaging (related to Flocking by Olfati-Saber, 2006). Are you familiar with this span of literature?``
>
> Thank you for pointing out this line of literature! There are some similarities but also noticeable differences between the two.
>
> As described in the paper you referenced [A], flocking could be achieved using a potential function to be specified between agents and using the averaged unnormalized gradients of these potentials as the update rule. When this potential function is piece wise linear in agents' positions, then the flocking algorithm update condenses to having a weighted averaging of directions. In [A], the author considers a potential function that is very close to a piecewise linear function of the distances between the agents (see Fig. 3 of [A]). Consequently, their update rule (averaged unnormalized gradients of the potential functions) boils down to averaging gradient directions. In contrast, DiBS does not assume any structure of the agent objectives (potential functions in flocking), and its updates will always be dictated by directions regardless of the agent objectives. Finally, we note that DiBS dynamically weights these directions according to the agents' respective distances to their optima.
>
> We are happy to updated our related works section to reflect this discussion.
>
> ``On line 274 - can you please clarify or cite a specific query complexity for estimating the normalized gradient?``
>
> This complexity varies with the method one wants to use. For example, ST-CR (reference [16] in our manuscript) guarantees that after $k$ comparisons to estimate a $n$ dimesnional normalized gradient, either the agent is at a near-optimal point (i.e., small gradient magnitude), or the angle between the true normalized gradient $\nabla \ell$ and the estimated normalized gradient $\nabla \tilde \ell$ is bounded as
> $$\angle(\nabla \tilde \ell, \nabla \ell) \leq \sin^{-1}(\sqrt{1- \frac{1}{2n}}^{\lfloor{\frac{k-n}{n-1}\rfloor}}).$$
>
> ``Can you comment on how theoretically accurate a direction oracle approach would be for computing DiBS? This question appears to be what is experimentally validated through Sec 4.2.``
>
> Thank you for this interesting question. Based on the context from previous question, we believe that the reviewer intends to ask "how theoretically accurate a *comparison* oracle approach would be for computing DiBS?" As we mentioned in the previous question, using the comparison oracle, the direction can be estimated to any desirable accuracy, and one may potentially show that the update direction of DiBS using the comparison oracle will have bounded error with respect to DiBS using the exact directions. However, this is non-trivial and providing a more formal theoretical statement to this end remains a direction for future work. we present empirical evidence of promising robustness properties of DiBS against such errors in Section 4.2, as the reviewer correctly points out.
>
> ``In Sec 4.2, can you please clarify how this example is a cooperative bargaining game? Whereas it was clear how the example in Sec 4.1 compared DiBS to NBS and KSBS, I'm not immediately seeing how you're computing DiBS, given the error on line 337.``
>
> This example in question refers to the mediated portfolio management experiment.
>
> **Purpose of Portfolio experiment.** As we mention above, the purpose of experiment in Section 4.2 is to see how reliably can DiBS solution be computed using only comparison oracles, i.e., using only $\mathcal O_{\mathbf{\ell}}^{\mathcal C, i}$ introduced in equation 3 to approximate $\mathcal O_{\mathbf{\ell}}^{\mathcal D, i}$ introduced in equation 1. The error metric mentioned in line 337 is used to measure the quality of solution produced by DiBS using only comparisons. In this example, we do not compare DiBS against NBS or KSBS, but demonstrate a practical implementation of DiBS using comparisons.
>
> **How is the portfolio experiment a bargaining game?** Each of the N investors is an agent. The statespace dimension is the number of stocks being considered, and each component of the state represents what fraction of the portfolio should be allocated to a particular stock. All agents pool their investments into a joint portfolio. The mediator (who managing the portfolio on behalf of investors) must conduct a bargaining process to find an portfolio allocation which is amenable to all agents, each of whom have a quadratic cost given by Markowitz portfolio theory (reference [13] in our paper), $\ell^i(x) = x^\top \Sigma^i x - \lambda^i \mu^{i\top} x$. The quantities $\Sigma^i, \mu^i$ are computed from real stock data taken over different time periods for each agent, reflecting differing market perspectives for each agent. The quantity $\lambda^i$ quantifies the risk-reward tradeoff coefficient for every agent. This is the set up for the bargaining game. We solve this game for 100 different randomly sampled time intervals of stock data and randomly sampled $\lambda^i$ for each agent, (leading to 100 $(\Sigma^i, \mu^i, \lambda^i)$ tuples) and each of these games (initialized at $x_0$) is solved by the DiBS algorithm in two ways
> * Directly using the true normalized gradients of the quadratic cost above in the DiBS updates (leading to solution $x^\dagger_{dir}$).
> * Using only comparisons to estimate the normalized gradients, using only $1, 10, 100, 1000, 10000$ comparisons per iteration per agent in the DiBS updates (leading to solutions $x^\dagger_{comp}$).
>
> Our aim is to see how different the true and estimated DiBS iterates are, thus the error metric on line 337 that the reviewer mentions is used. It shows the relative error, given by $\frac{\|x^\dagger_{dir} - x^\dagger_{comp}\|}{\|x^\dagger_{dir} - x_0\|}.$
> The numerator measures how close the comparison based solution is to the direction based solution. The denominator is acting as a normalizer to account for the size of state space (and also to account for the fact that $x_0$ changes across initializations). The statistics are reported over the 100 game initializations, each $x_0$ being different.
> We hope this clarifies how the protfolio experiment in Section 4.2 is a bargaining game. We are happy to provide any further clarifications required.
>
> ``Extensions - I'm wondering what other types of problems you think this technique could be applied to beyond cooperative bargaining.``
>
> Thanks for this interesting question. One interesting direction we think can be applying such comparison/oracle based techniques for the flocking problem that you mentioned. It will be very interesting to see if in a decentralized setting, self interested agents can employ such oracles as an alternative to predefined potential functions.
>
>
>
>
>
> ``Minor comments pointed by reviewer``
> * **Line 114:** thank you, we will add appropriate citation for the NBS iterates converging to desirable point.
> * **Please elaborate on how SignSGD can be used to compute the most preferred states on Line 218:** SignSGD [B] (and related inspired algorithms) use a sampling scheme to estimate a normalized gradient $\tilde g$, and then use a gradient descent like procedure with the signum values of the normalized gradient estimates, $x \leftarrow x - \eta \cdot\text{sign}(\tilde g)$. They utilize the fact that for smooth functions, the signum transformed gradient will only travel in the wrong direction for a component, if that component is already close to the optimal value. As such, these methods converge to preferred states with appropriately chosen step sizes $\eta$. For example, one method inspired from SignSGD, the Sign-OPT algorithm, (reference [5] in our paper) samples random directions $u_1, ..., u_Q$ and queries whether the function value increases or decreases along each direction via the comparison oracle. The sign information is used to construct an estimate of the gradient direction of a function $\ell^i$, $\hat g^i$ as
> $\hat{g}^i = \frac{1}{Q} \sum_{q=1}^Q \mathcal O^{\mathcal O, i}_{\mathbf \ell}(x + \epsilon u_q, x) \cdot u_q,$ where $Q$ is the number of comparisons and $\epsilon>0$ is a small smoothing parameter.
> This gradient estimate is then used to perform a gradient descent type update to estimate $x^{*,i}$ as $x \leftarrow x - \eta \hat{g}^i$. Despite the lack of gradient magnitude information, the method achieves a convergence rate of $\mathbb{E}[||\nabla \ell^i(x)||_2] = \mathcal{O}\left( \frac{\sqrt{n}}{\sqrt{T}} + \frac{n}{\sqrt{Q}} \right)$ where $n$ is the state dimension, $T$ is the number of iterations, and $Q$ is the number of sampled directions/comparisons made per iteration. We will update the manuscript to reflect this discussion.
> * **Why is proof of Theorem 1 a proof "sketch" (line 452)?** The sketch consists of only the first paragraph (lines 452 - 456). The rest is the full proof. We will modify our manuscript to make this more clear. Thank you for pointing this out.
>
>
> [A] Olfati-Saber, Reza. "Flocking for multi-agent dynamic systems: Algorithms and theory." IEEE Transactions on automatic control 51.3 (2006): 401-420.
>
> [B] Bernstein, Jeremy, et al. "signSGD: Compressed optimisation for non-convex problems." International conference on machine learning. PMLR, 2018.

---

> > ### Comment · Reviewer_XLAd · 2025-08-05
> >
> > Thank you for your rebuttal to my and other reviewers' comments. This helps clear up my confusion, and it would help the paper by including some of this discussion into the final paper. I will keep my score positively as-is.

---

### Official Review · Reviewer_PReA · 2025-06-30

**Clarity:** 3
**Significance:** 3
**Originality:** 4
**Rating:** 5
**Confidence:** 3

**Summary:**

This paper studies cooperative bargaining games where mediators lack access to agents' utility values but only have query access to a direction oracle that returns each agent’s normalized utility gradient which is their most preferred direction in the decision space. Existing solutions like Nash (NBS) and Kalai-Smorodinsky (KSBS) bargaining require utility values or gradients and fail under nonaffine utility transformations. The authors prove that no algorithm using only direction oracles can recover NBS or KSBS.  The authors propose Direction-based Bargaining Solution (DiBS) in which mediator iteratively updates the state using agent’s preferred state which is obtained via precomputation or oracles. This weights each agent’s direction by their distance to their ideal state by prioritizing agents farther from their goals and uses only direction oracles and preferred states. Under convexity and smoothness assumptions, DiBS converges globally to Pareto stationary points. DiBS is robust to strictly increasing nonaffine utility transformations, which is a preferred invariance property not satisfying by NBS/KSBS.  It also satisfies Pareto stationarity, symmetry, and independence of irrelevant alternatives under uniqueness assumption.  In practice, direction oracles can be estimated from binary comparison oracles, like agents answer yes or no to state proposals. This is validated in two example domains of nonconvex multi-agent formation and stock portfolio allocation with diverse investor preferences.  Overall, this paper is a significant contribution to addressing problems related to human-AI collaboration, privacy-aware systems, or preference learning, etc. where utilities are hard to specify.

**Questions:**

- The prosed method requires knowing agents’ ideal states. How robust is it to approximate states and noisy comparisons? The authors mention pre-computation, but I think it might be useful to have some sensitivity analysis.
- Classical bargaining uses disagreement points to anchor negotiations which the proposed method omits. Does it weaken fairness in asymmetric scenarios?
- How does the proposed method compare to simple heuristics, like weighted averaging of directions, in speed, fairness, robustness?
- Could the proposed method extend to settings where agents’ goals evolve during bargaining?

**Ethical Concerns:**

["NO or VERY MINOR ethics concerns only"]

**Final Justification:**

This paper presents strong theoretical results. The authors have addressed all my questions during the rebuttal.

**Limitations:**

Yes

**Quality:**

4

**Strengths And Weaknesses:**

Strengths:
- This work has significant novelty. Prior works have used directional information but are limited to two agents or produces unfair outcomes.
The proposed method seems to be the first multi-agent bargaining solution with direction oracles that guarantees fairness and axiomatic properties. The focus on nonaffine invariance is also new.
- The theoretical proofs seem sound. It is intuitive that DiBS fixed points correspond to Pareto stationary points, and invariance to nonaffine transformations holds by construction.
- Experiments show that the proposed method achieves balanced solutions comparable to NBS and KSBS, it is indeed robust to nonaffine transformation and works well with comparison based gradient estimates.

Weaknesses:
- The theoretical proofs of global convergence assume strong convexity, while experiments show that event formation converges even for non-convex cost. However, this is standard for theoretical analysis in this topic. The model also assumes static preferences.
- Experiments lack comparison with simple heuristics, like weighted averaging of directions.

---

> ### Author Rebuttal · Authors · 2025-07-31
>
> Thank you for your review and comments! Please find our responses below.
>
> ``The prosed method requires knowing agents’ ideal states. How robust is it to approximate states and noisy comparisons? The authors mention pre-computation, but I think it might be useful to have some sensitivity analysis.``
>
> Thank you for this interesting question! We interpret your question as having two parts:
> 1. **How accurately can one calculate preferred states using noisy comparisons/ noisy normalized gradients estimates?**
>     We remark that prior works in single agent optimization have a rich variety of methods which can be used to estimate preferred states with noisy comparisons/normalized gradients.
> * In case of noisy comparisons, one can use the method in [A] to find the preferred state up to desired accuracy, with a known query complexity of achieving a state within a certified error tolerance (see Theorem 2, [A]).
> * In case of accurate comparisons but noisy normalized gradients, one can use methods which work with noisy gradient directions like Sign-OPT [B] or Sign-SGD [C], which conduct a gradient descent like procedure with the signum values of the direction oracle and show that these methods converge to preferred states with suitable step size selection. They utilize the fact that for smooth functions, the signum transformed gradient will only travel in the wrong direction for a component, if that component is already close to the optimal value. For example, with $Q$ comparisons per iteration, after $T$ iterations, even by using imperfect and noisy normalized gradient estimates, SignOPT yields a state $x$ such that $$\mathbb{E}[\|\nabla \ell^i(x)\|_2] = \mathcal{O}\left( \frac{\sqrt{n}}{\sqrt{T}} + \frac{n}{\sqrt{Q}} \right).$$ Another approach, which uses a cutting plane/ellipsoid based method to obtain preferred states is [D], which iteratively identifies and rejects certifiably sub optimal regions of the state space, to find a preferred state upto desired accuracy.
> 2. **How robust is our method DiBS to noisy estimates of preferred states $x^{*,i}$?**
> Thank you for this interesting question. We tried to conduct some analysis for this question, but our efforts have not yet led to a meaningful result, and exact theoretical robustness guarantees remain an interesting direction for future work. However, we have conducted some empirical studies on the nonconvex multi-agent formation experiment of Section 4.1 (
> where due to the symmetry of the problem, noisy agent preferred states can substantially change the problem geometry). We noticed that qualitatively the formations resulting from the true and noisy preferred states are visually quite similar and, as expected, the noisy state formation gets closer to the original solution as the approximation gets better and noise decreases. We are happy to share the quantitative results of this empirical study, and also revise our manuscript to include this study.
>
>
>
> ``Classical bargaining uses disagreement points to anchor negotiations which the proposed method omits. Does it weaken fairness in asymmetric scenarios?``
>
> The reviewer is correct in pointing out that our proposed approach does not consider improvement over a disagreement point. Instead, the notion of fairness we consider is that the bargaining solution not be too close to any one particular agent's preferred state -- this means in scenarios where one agent starts close to their preferred state (i.e., the disagreement point is close to their preferred state), our approach specifically de-emphasizes their contribution to the update, and drives the update to a less favourable state for this agent, while making the solution more favourable for other agents. We believe that we can adjust the DiBS update rule to account for disagreement points: either by introducing a term that scales directions based on how close the disagreement point is to each agent's optimum, or by projecting the DiBS update into the space of mutually improving directions. We believe this can be an exciting direction for future work, and thank the reviewer for raising this comment!
>
> ``How does the proposed method compare to simple heuristics, like weighted averaging of directions, in speed, fairness, robustness?``
>
> *We note that the heuristic of a simple weighted averaging scheme where the weights are constant does not act as a valid bargaining solution, because the iterates of such a simple heuristic need not converge.* As an example, consider the bargaining game $\mathcal{B}_{[0,1]}([x^2, (x-1)^2], [1,1])$ given in Example 2 of our paper, but instead of using equation (2), consider using a weighted averaging update rule which weighs both the gradients by $a, b$ respectively, where $a\neq 0, b\neq 0, a+b = 1$. Let $a>\frac{1}{2}$ without loss of generality, then we have that the iterates of such an update rule do not have equilibrium for any $x\in[0,1]$, and the updates never converge, because the iterates either want to move to 0 (agent 1's preferred state) for $x\in(0,1]$, or to 2 (agent 1's preferred state) when $x=0$. We hope this also clarifies why we do not have experimental comparisons with such simple heuristics, which was brought up by the reviewer in the weaknesses section.
> Thus, it is not possible to compare DiBS on robustness or fairness against such weighted average schemes.
>
> Further, we remark that DiBS can be viewed as a non-trivial weighted averaging of directions scheme, but where the weights evolve dynamically across iterations. DiBS provides one way to construct weights using only information available through comparisons.
>
> Regarding the speed of DiBS, assuming that preferred states $x^{ *,i}$ are found as a precomputation step, calculating the weights $||x-x^{ *,i}||$  involve only 1 vector subtraction and 1 norm calculation, which leads to an efficient linear time complexity of $\mathcal{O}(nN)$ per iteration, where $n$ is the state dimension and $N$ is the number of agents. This is in addition to the complexity of calculating the normalized gradients, which will also be incurred by any algorithm which utilizes normalized gradients.
>
> ``Could the proposed method extend to settings where agents’ goals evolve during bargaining?``
>
> Thank you for this interesting question. As of now, the proposed approach only considers the setting where agent costs $\ell^i$ and preferred states $x^{*,i}$ are fixed, and do not evolve with time. Extending the approach to accomodate evolving agents' goals is an direction for future work, and we hypothesize this might consist of constructing updates which converge on a faster time scale than the timescale on which the agent goals evolve.
>
> [A] Jamieson, Kevin G., Robert Nowak, and Ben Recht. "Query complexity of derivative-free optimization." Advances in Neural Information Processing Systems 25 (2012).
>
> [B] Cheng, M., Singh, S., Chen, P. H., Chen, P. Y., Liu, S., & Hsieh, C. J. Sign-OPT: A Query-Efficient Hard-label Adversarial Attack. In International Conference on Learning Representations (2020).
>
> [C] Bernstein, Jeremy, et al. "signSGD: Compressed optimisation for non-convex problems." International conference on machine learning. PMLR, 2018.
>
> [D] Karabag, Mustafa O., Cyrus Neary, and Ufuk Topcu. "Smooth convex optimization using sub-zeroth-order oracles." Proceedings of the AAAI Conference on Artificial Intelligence. Vol. 35. No. 5. 2021.

---

> ### Comment · Reviewer_PReA · 2025-08-03
> **Rebuttal discussion**
>
> I thank the authors for the rebuttal. All my concerns are addressed. I am happy to keep my score as 5 (Accept), as I think I am not knowledgeable enough about the problem and the full technical details to recommend a 6 (Strong Accept) for the paper.

---

### Official Review · Reviewer_S2R2 · 2025-07-02

**Clarity:** 3
**Significance:** 3
**Originality:** 3
**Rating:** 4
**Confidence:** 2

**Summary:**

The paper proposes a solution concept for bargaining that is invariant under nonmonotone utility transformations, and show that it enjoys desirable properties. Furthermore, existing concepts do not satisfy all these desiderata simultaneously. On the algorithmic side, the authors show dynamics that asymptotically converge to the concept.

**Questions:**

1. Could you please let me know if I'm not on the right track with my confusion around L38 described above?
2. What can be said about non-asymptotic guarantees, especially under discretization?

**Ethical Concerns:**

["NO or VERY MINOR ethics concerns only"]

**Final Justification:**

The author's rebuttal has helped clarify a point of confusion regarding line 38, as mentioned above. The authors have shown goodwill to fix the typos and minor issues I raised. I maintain a positive opinion of the paper. As mentioned in the review, I am not an expert on bargaining, and my review focused mostly on the quality of writing and clarity.

**Limitations:**

Yes.

**Paper Formatting Concerns:**

No concerns about formatting

**Quality:**

3

**Strengths And Weaknesses:**

I am not an expert on bargaining, so I will focus mostly on the quality of writing and clarity.

Overall, I found the paper well-written. However, as a non-expert on bargaining, I would have certainly appreciated a bit more information at the very beginning of the paper, if anything to remind the reader of the main challenges of bargaining and the settings it tries to capture.

On L38, I would have appreciated a more nuanced discussion about how reasonable it is to be unsatisfied with the existing approaches' lack of nonaffine invariance. Under a nonaffine transformation, the relative amount with which an outcome is better than another is not preserved. So, it seems quite unclear to me what, abstractly, we would want to require out of a good bargaining approach. The authors liquidate this quite fast, by saying that "despite utilities being incomparable due to different nonaffine scalings, the notion of the agents’ most preferred directions remains intact." I don't disagree, but why is focusing on direction, instead of scaling, reasonable? And under this relaxed goal, is it obvious that prior approaches behave unreasonably? Up until that point, in the mind of the reader, the only negative regarding the prior approaches is just a lack of affine invariance and the possibility of them "favor(ing) one agent disproportionately". Nothing has been said about their inability to try to move, in some sense, in the direction most preferred by each player.

I found the overall discussion on relations with the existing literature well done.

Minor comments:
- Reference [2]: I think the capitalization of "Yahoo! Finance" and "API" is probably incorrect
- Reference [4]: I don't speak German, but I believe they capitalize nouns. If so, it might be worth double checking that the capitalization used is correct
- Citations: A quite large number of references point to arXiv. Perhaps this is correct, but I find it somehow unusual. Could you please double check that you are citing the most relevant version (for example, if a paper has been accepted in proceedings or into a journal, please provide a citation to that published version, or a note on why the arXiv version is the correct version you refer to).

---

> ### Author Rebuttal · Authors · 2025-07-31
>
> Thank you for your review and comments! Please find our responses below.
>
> ``Could you please let me know if I'm not on the right track with my confusion around L38 described above?
> ``
>
> We thank the reviewer for bringing up this point and allowing us to clarify. The reason why it is desirable for a bargaining algorithm (and solution) to be invariant to nonaffine scalings of agents' objectives is because it destroys any incentive for those agents to exaggerate the degree to which they prefer one outcome over another. However, the chain rule reveals that - for any (potentially unknown) nonaffine but monotone scaling function $\sigma: \mathbb{R} \to \mathbb{R}$ and objective function $\ell^i$ for agent $i$ - $\nabla \sigma(\ell^i(x)) = \sigma'(\ell^i(x)) \nabla \ell^i(x)$ which implies that $\nabla \sigma(\ell^i(x))$ is parallel to $\nabla \ell^i(x)$ (and thus, while the normalized gradients are equal, algorithms like NBS and KSBS which use the unnormalized gradient now get changed values). This argument is currently stated in the proof of Proposition 1 (line 446 in the Appendix), but we will factor it into a standalone Lemma and reference it in the main text in order to improve clarity.
>
> Finally, we demonstrate in Example 1 and Proposition 1 that existing approaches such as NBS and KSBS can fail to recover the same solution when they lack access to the true (pre-transformation, un-exaggerated) objective functions. By exaggerating its reported objective values, therefore, one agent can easily bias the result. On the other hand, since the ordinal comparisons between points remain the same, DiBS is unaffected by this type of misreporting.
>
> We believe that the reviewer may also be suggesting that existing bargaining approaches could, in principle, handle non-affine transformations if they knew the transformation itself and could apply its inverse to recover the underlying objectives. However, the key feature of DiBS is that it assumes the mediator only has access to a direction oracle for each agent's objective function. In particular, the mediator does not observe the actual objective values and is agnostic to possible (non-affine) objective transformations. Consequently, DiBS identifies the same solution for any monotone transformation.
>
> ``What can be said about non-asymptotic guarantees, especially under discretization?``
>
> Thank you for bringing this up. We agree that it would be useful to understand DiBS's nonasymptotic convergence behavior. Unfortunately, our analysis to date has not led to any meaningful results. We certainly intend to continue, but at this point we must defer that effort for future work.
>
> ``Other minor comments that the reviewer brings up:``
> * On capitalization in references: Thank you for pointing this out, we will edit references [2] and [4] appropriately.
> * Arxiv citations: Thank you for this observation, we will replace the citations for those papers who have published versions available.

---

> > ### Comment · Reviewer_S2R2 · 2025-08-07
> >
> > Dear authors, thank you for your response and clarification. I maintain a positive opinion of the paper.

---

### Official Review · Reviewer_Rh96 · 2025-07-04

**Clarity:** 3
**Significance:** 3
**Originality:** 3
**Rating:** 5
**Confidence:** 2

**Summary:**

The paper focuses on cooperative bargain games, when utility functions are not available, or different agents' utilities are not comparable. The contributions include: 1) showing that no algorithms with only direction oracle access can find Nash or Kalai-Smorodinsky bargaining solutions for all bargaining games; 2) proposing Direction-based Bargaining Solution (DiBS), an interative algorithm that incorporates agents' distance from preferred state, using only direction oracles. 3) showing that fixed points of DiBS exists and are Pareto stationary.

**Questions:**

* You mentioned that naive direction oracle-based bargaining can lead to unfair solutions. In experiments, DiBS seems to have more balanced solutions. Is there any theoretical explanation for this?
* How to find the preferred state with direction oracle access? Intuitively, you could follow the oracle directions. But the algorithm is nonstandard as you don't have the full gradient, so it'll be helpful to explicitly state the algorithm.

**Ethical Concerns:**

["NO or VERY MINOR ethics concerns only"]

**Final Justification:**

I'm happy to keep the score, which is already 5. However, I have a low confidence score for my assessment as it is not my area of expertise.

**Limitations:**

Yes

**Paper Formatting Concerns:**

No formatting concerns.

**Quality:**

3

**Strengths And Weaknesses:**

* Identifies a practical problem of cooperative bargaining games without utility access, which has wide applications with AI proxy agents.
* Proposes DiBS, which provably achieves Pareto stationarity under standard assumptions
* The paper is well-writen, and has strong technical contributions.

---

> ### Author Rebuttal · Authors · 2025-07-31
>
> Thank you for your comments and review! Please find our responses below.
>
> ``You mentioned that naive direction oracle-based bargaining can lead to unfair solutions. In experiments, DiBS seems to have more balanced solutions. Is there any theoretical explanation for this?``
>
> The naive direction oracle based bargaining method given in equation (2) indeed can result in unfair solutions (as argued in example 2). On the other hand, as correctly observed by the reviewer, our method DiBS indeed results in balanced solutions. The *theoretical explanation* behind this, is that the DiBS algorithm does not have a simple/naive aggregation of direction oracles, instead DiBS can be interpreted as a weighted aggregation of directions, where weights for all agents evolve dynamically throughout the bargaining iterations (given at line 207, Section 3). These weights evolve in a manner which ensures fair/balanced solutions, with agents far away from their desired states having more priority. In comparison, a simple aggregation of directions such as given by equation (2) represent a weighting scheme which does not evolve in a manner which ensures balanced solutions.
>
>
>
> ``How to find the preferred state with direction oracle access? Intuitively, you could follow the oracle directions. But the algorithm is nonstandard as you don't have the full gradient, so it'll be helpful to explicitly state the algorithm.``
>
> As we mention in Section 3.2 (lines 269-279), there are various different methods in existing optimization literature which can be used to find preferred states from just direction oracle, some recent and relevant methods being the references [5,9,12,22,32] in our manuscript. A popular method is to conduct a gradient descent like procedure with the *signum values of the direction oracle*, for example in SignSGD [A], or Sign-OPT [5], having an update of the form $x \leftarrow x - \eta\cdot \textrm{sign}(g)$, where $g$ is the direction oracle output. These papers show that with a properly chosen step size $\eta$, such methods enjoy convergence to agents' preferred states. These methods utilize the fact that for smooth functions, the signum transformed gradient will only travel in the wrong direction for a component, if that component is already close to the optimal value.
>
> We are happy to appropriately modify our manuscript to reflect this discussion, add prior work's algorithm in the appendix and mention the precise convergence rates for these prior methods.
>
> [A] Bernstein, Jeremy, et al. "signSGD: Compressed optimisation for non-convex problems." International conference on machine learning. PMLR, 2018.

---

> > ### Comment · Reviewer_Rh96 · 2025-08-03
> >
> > Thank you for your response. I'm happy to keep my score at 5.

---

### Note · Authors · 2025-08-13

We thank all reviewers and chairs for their efforts. Based on the reviewers' final responses, we believe that we addressed all reviewer queries. We will make appropriate modifications to the manuscript to reflect rebuttal discussions and suggestions made by reviewers.

---

### Decision · Program_Chairs · 2025-09-17

**Decision:**

Accept (poster)

**Comment:**

This submission considers cooperative bargain games. In particular, it studies the case when utility functions are not available, or different agents' utilities are not comparable. The following main contributions are contained in the submission:
1) It is shown that no algorithm with only direction oracle access can find Nash or Kalai-Smorodinsky bargaining solutions for all bargaining games.
2) The authors propose the concept of a Direction-based Bargaining Solution (DiBS), an iterative algorithm which uses only direction oracles, and reasons about every agent’s distance from their preferred state throughout the bargaining process.
3) It is shown that fixed points of DiBS exist and are Pareto stationary (under strong convexity and smoothness assumptions).

All reviewers are very positive about this submission, and all remaining questions could be clarified in the discussion such that no severe weaknesses remain. The strengths are the following: The studied problem is of practical interest in several settings, including, e.g., human-AI interactions. The technical contribution has significant novelty: While prior works have used directional information limited to two agents or produces unfair outcomes, the proposed method seems to be the first multi-agent bargaining solution with direction oracles that guarantees fairness and axiomatic properties. The paper is very well-written. Furthermore, the experiments show that the proposed method achieves balanced solutions comparable to NBS and KSBS, it is indeed robust to non-affine transformation and works well with comparison based gradient estimates.